

# Carbon isotopes and Pa/Th response to forced circulation changes: a model perspective

Lise Missiaen[1,2], Nathaelle Bouttes[1], Didier M. Roche[1,3], Jean-Claude Dutay[1], Aurélien Quiquet[1,4], Claire Waelbroeck[1],
Sylvain Pichat[5,6], Jean-Yves Peterschmitt[1]

[1]Laboratoire des Sciences du Climat et de l'Environnement, LSCE/IPSL, CEA-CNRS-UVSQ-Université Paris-Saclay, F-91198 Gif-sur-Yvette, France.
[2]Climate Change Research Centre, University of New South Wales, Sydney, Australia
[3]Vrije Universiteit Amsterdam, Faculty of Science, Cluster Earth and Climate, de Boelelaan 1085, 1081HV Amsterdam, The Netherlands
[4]Institut Louis Bachelier, Chair Energy and Prosperity, Paris, 75002, France
[5]Université de Lyon, ENS de Lyon, Laboratoire de Géologie de Lyon (LGL-TPE), F-69007 Lyon, France.
[6]Climate Geochemistry Department, Max Planck Institute for Chemistry, Mainz, Germany.

*Correspondence to*: Lise Missiaen (l.missiaen@unsw.edu.au)

**Abstract.** Understanding the ocean circulation changes associated with last glacial abrupt climate events is key to better assess climate variability and understand its different natural modes. Sedimentary Pa/Th, benthic $\delta^{13}$C and $\Delta^{14}$C are common proxies used to reconstruct past circulation flow rate and ventilation. To overcome the limitations of each proxy taken separately, a better approach is to produce multi-proxy measurements on a single sediment core. Yet, different proxies can provide conflicting information about past ocean circulation. Thus, modelling them in a consistent physical framework has become necessary to assess the geographical pattern, the timing and sequence of the multi-proxy response to abrupt circulation changes.

We have implemented a representation of the $^{231}$Pa and $^{230}$Th tracers into the model of intermediate complexity iLOVECLIM, which already included $\delta^{13}$C and $\Delta^{14}$C. We have further evaluated the response of these three ocean circulation proxies to a classical abrupt circulation reduction obtained by freshwater addition in the Nordic seas under preindustrial boundary conditions. Without a priori guess, the proxy response is shown to cluster in modes that resemble the modern Atlantic water masses. The clearest and most coherent response is obtained in the deep (> 2,000m) Northwest Atlantic, where $\delta^{13}$C and $\Delta^{14}$C significantly decrease while Pa/Th increases. This is consistent with observational data across millennial scale events of the last glacial. Interestingly, while in marine records, except in rare instances, the phase relationship between these proxies remains unclear due to large dating uncertainties, in the model the bottom water carbon isotopes ($\delta^{13}$C and $\Delta^{14}$C) response lags the sedimentary Pa/Th response by a few hundred years.

## 1 Introduction

Understanding rapid climate changes is key to understand the different natural modes of ocean circulation as they provide information about internal climate variability and climate sensitivity to perturbations. Indeed, during the last glacial, rapid and high-amplitude atmospheric temperature changes of about 8 to 15°C in less than 300 years are associated with only small changes in radiative forcing (Kindler et al., 2014). Changes in the Atlantic Meridional Overturning Circulation (AMOC) strength are thought to be one of the main drivers of these abrupt climate events (see (Lynch-Stieglitz, 2017) for a review)), but the underlying mechanisms remain elusive. Available AMOC records based on observations only cover a few decades, thus identifying the processes controlling AMOC changes can only be achieved from the study of long-term variations (Smeed et al., 2014), which relies on the analysis of indirect evidence (paleoproxies).

To date, among the numerous tracers available, the most valuable ones to reconstruct and quantify past circulation patterns and water mass flow are the sedimentary ($^{231}$Pa$_{xs,0}$/$^{230}$Th$_{xs,0}$) ratio (hereafter Pa/Th) and dissolved inorganic carbon isotopes ($\delta^{13}$C, $\Delta^{14}$C).



The Pa/Th is the ratio of the $^{231}$Pa and $^{230}$Th activities (decays per time unit) at the deposition time that are derived from the water column scavenging. The Pa/Th ratio can be used as a kinematic circulation proxy (François, 2007; McManus et al., 2004). In short, $^{231}$Pa and $^{230}$Th are homogeneously produced in the water column at known rates with a known production ratio of 0.093 and are then transferred to the underlying sediments by particle scavenging (see (François, 2007) for a review). The two

isotopes have different residence times in the water column (50-200 years for $^{231}$Pa and 10-40 years for $^{230}$Th (Henderson and Anderson, 2003)). Thus, while $^{230}$Th is rapidly transferred to the sediment, $^{231}$Pa can be partly transported by water mass advection along the large-scale ocean circulation. Consequently, the sedimentary Pa/Th activity ratio can be used as a proxy of water-mass advection rate. In the Atlantic, low sedimentary Pa/Th ratios (*e.g.* 0.04 for the modern North Atlantic (Yu et al., 1996)) are diagnostic of an active overturning, while Pa/Th ratios close or equal to the production ratio indicate a sluggish water

mass or a marked overturning circulation slowdown (*e.g.* (Böhm et al., 2015; François, 2007; McManus et al., 2004; Waelbroeck et al., 2018)). Yet, Pa and Th scavenging to the sediment is sensitive to changes in vertical particle flux and composition; hence the sedimentary Pa/Th circulation signal could be partly impaired by a particle-related signal (*e.g.* (Chase et al., 2002, 2003; Lippold et al., 2009)). In addition, sedimentary Pa/Th is derived from bulk sediment measurements and requires the estimation of the contributions of the detrital and authigenic fractions to the $^{231}$Pa and $^{230}$Th budgets. We have recently shown that this

estimation can lead to significant uncertainties on the reconstructed patterns and amplitudes of the Pa/Th signal, especially in locations characterized by high terrigenous inputs (Missiaen et al., 2018). These potential caveats need to be tested and could complicate the evaluation of past circulation strength changes from Pa/Th measurements.

The carbon isotopes measured in foraminifer shells reflect the carbon isotopic content of the water mass in which they form (Stuiver and Polach, 1977) and provide information about past water mass ventilation, in other words, the time elapsed since

the tracked deep-water parcel has been isolated from the surface (*e.g.* (Lynch-Stieglitz et al., 2014; Skinner et al., 2014)). More precisely, the water mass $\delta^{13}$C signature depends on biological and physical processes. Carbon is exchanged between the surface waters and the atmosphere. In the surface waters, carbon is incorporated into the organic matter through biologic productivity. Both air-sea exchange and biological activity are responsible for isotopic fractionation between $^{12}$C and $^{13}$C (Siegenthaler and Münnich, 1981). Consequently, the surface waters $\delta^{13}$C signature varies with air-sea exchange efficiency and

biological activity intensity. At depth, remineralization of organic matter releases $^{13}$C to the water parcels, which are then mixed through large-scale ocean circulation. In the modern ocean, the global $\delta^{13}$C distribution depicts a tight relation between the apparent oxygen utilization and the $\delta^{13}$C signature of a given water mass (Eide et al., 2017). This observation lends support to the use of the $\delta^{13}$C of benthic foraminifers as a proxy for ocean oxygen content and ventilation (Duplessy et al., 1988). However, benthic $\delta^{13}$C does not solely record deep ventilation changes. As mentioned above, the $\delta^{13}$C signature of a deep-water mass

depends on several processes: its value before the water left the surface mixed layer, the intensity of the biological activity in the mixed layer, the remineralization intensity at depth and finally the circulation path and strength. Thus, benthic $\delta^{13}$C records multiple processes, which complicates its interpretation in terms of past deep ocean ventilation and circulation changes.

Radiocarbon is produced in the upper atmosphere and enters into the ocean via air-sea exchange with surface waters. As soon as a water parcel is isolated from the surface, its $^{14}$C content starts to decrease exponentially with time due to radioactive decay

(half-life of $^{14}$C = 5,730 ± 40 years (Godwin, 1962)). Thus, by determining the $^{14}$C age of benthic foraminifer samples of independently known calendar age, one can reconstruct past ocean ventilation, (*e.g.* (Skinner and Shackleton, 2004; Thornalley et al., 2015)). However, the interpretation of a water mass radiocarbon age is complicated by temporal variations in $^{14}$C production in the upper atmosphere and air-sea exchange efficiency. Indeed, since radiocarbon is only produced in the atmosphere and transferred to the ocean via air-sea exchange, the surface waters have an older radiocarbon age than the

contemporaneous atmosphere. This radiocarbon age difference between the surface waters and the atmosphere (termed the surface reservoir age) can vary with space and time according to variations in air-sea exchanges efficiency, especially in the North Atlantic region (see (Bard et al., 1994; Bondevik et al., 2006; Thornalley et al., 2011; Waelbroeck et al., 2001)). Those



variations are still poorly constrained, and complicate the interpretation of deep water radiocarbon content (expressed as $\Delta^{14}C$ in ‰) in terms of past ocean ventilation and circulation changes (*e.g.* (Adkins and Boyle, 1997)).

Pa/Th, benthic $\delta^{13}C$ and $\Delta^{14}C$ can provide useful information about past ocean circulation flow rate, geometry and ventilation. Yet, as highlighted above, each proxy has its own caveats. To overcome the limitation of each proxy taken separately and gather

more detailed information about past ocean circulation, paleoceanographers started to conduct multi-proxy studies, producing different proxy records on the same sedimentary archive. Moreover, this approach also enables one to investigate phase relationships between the different proxies (Burckel et al., 2015; Waelbroeck et al., 2018). Indeed, current dating uncertainties in marine cores are generally less than 150 y between 0 and 11 ky cal BP (ka), increasing to ~400 y between 11 and 30 ka and ranging from ~600 to 1100y between 30 and 40 ka (Waelbroeck et al, submitted). Such uncertainties usually prevent the

inference of phase relationships between proxy records from different marine cores, hence limiting the benefit of the multi-proxy approach.

Different proxies can bring conflicting information about past ocean circulation and reconstructing basin-scale ocean water-mass reorganization requires to compile proxy records from different locations. Despite some recent paleoproxy compilation efforts (Lynch-Stieglitz et al., 2014; Ng et al., 2018; Zhao et al., 2018), the amount of data available remains too sparse to

constrain the state and evolution of the ocean circulation in 3-D across abrupt climate events. One way to better understand existing paleo-records and overcome the scarcity of paleo-data is to use climate models able to simulate the evolution of different proxies in a consistent physical framework. Such work could help to explain why some events are not recorded or recorded differently at a given location. Climate models are also useful as they enable to analyze the spatial and temporal response of a proxy to changes of ocean circulation.

Because of its key role in the climate system, the carbon cycle has been modelled and heavily studied in the past decades (Friedlingstein et al., 2006; Orr et al., 2001). More recent studies simulate the evolution of carbon isotopes, and in particular $^{13}C$ and $^{14}C$, under pre-industrial or glacial conditions (Bouttes et al., 2015; Brovkin et al., 2002; Menviel et al., 2017; Tschumi et al., 2011). The $^{231}Pa$ and $^{230}Th$ tracers have also been implemented in climate models. The simplest approaches used 2D models (Luo et al., 2009; Marchal et al., 2000) or 3D models but contained oversimplifications, notably in the particles

representations (Siddall et al., 2005, 2007). Latest published developments focused on an improved representation of particle fluxes and scavenging scheme (van Hulten et al., 2018; Rempfer et al., 2017). However, these recent developments either suffer from the coarse resolution of the ocean model (Rempfer et al., 2017) which contains only 36 x 36 grid cells (latitude-longitude), or conversely cannot simulate 1,000 years in reasonable computation time (van Hulten et al., 2018). To our knowledge, there has not been any study so far that considered the geographical distribution and the temporal evolution of combined proxies such

as Pa/Th, $\delta^{13}C$ and $\Delta^{14}C$.

We further investigate the spatial and temporal structure of multi-proxy response to an abrupt circulation change with a climate model. For that purpose, we have implemented the production and scavenging processes of $^{231}Pa$ and $^{230}Th$ in the climate model of intermediate complexity iLOVECLIM. To date, the model is able to simulate the evolution of three ocean circulation proxies: Pa/Th, $\delta^{13}C$ and $\Delta^{14}C$. We also developed an analysis method that gives information about the magnitude and timing of the

proxy response in the Atlantic Ocean. In this study, we address the following questions. 1) What is the response of each proxy to an imposed circulation change? 2) What is the timing and sequence of the proxy responses? How do they vary with regions and water depth in the Atlantic Ocean? 3) How can the modelled multi-proxy response help to interpret the paleoproxy records?

## 2 Material and methods

### 2.1 Model description and developments

We use the Earth system model of intermediate complexity iLOVECLIM, which is a code fork of the LOVECLIM model (Goosse et al., 2010). It includes a representation of the atmosphere, ocean, sea ice, terrestrial biosphere vegetation, as well as





the carbon cycle. The ocean component (CLIO) consists of a free-surface primitive equation ocean model (3° x 3° horizontal grid, 20 depths layers) coupled to a dynamic-thermodynamic sea-ice model. iLOVECLIM includes a land vegetation module (VECODE) (Brovkin et al., 1997) and a marine carbon cycle model (Bouttes et al., 2015) both computing the evolution of $^{13}$C and $^{14}$C. Previous work has shown that the simulated oceanic $\delta^{13}$C and $\Delta^{14}$C distribution is in reasonable agreement with

observations and with available GCM simulations (Bouttes et al., 2015). We have implemented $^{231}$Pa and $^{230}$Th in the iLOVECLIM model following the approach of (Rempfer et al., 2017). In the model, $^{231}$Pa and $^{230}$Th are homogeneously produced in the ocean by radioactive decay of their respective uranium parents. They are removed from the water column by adsorption on settling particles (reversible scavenging). Their radioactive decay is also taken into account. In its current version, iLOVECLIM does not explicitly simulate the biogeochemical cycle of biogenic silica (opal), which is thought to be an important

scavenger for $^{231}$Pa (e.g. (Chase et al., 2002; Kretschmer et al., 2010)). Therefore, we used prescribed and constant particles concentration fields obtained running the PISCES biogeochemical model coupled to NEMO ocean model (van Hulten et al., 2018). Previous work has shown that these particle fields are reasonably consistent with present day observations (van Hulten et al., 2018). We considered one particle size with a unique sedimentation speed and three particles types (CaCO$_3$, POC and biogenic silica). The conservation equations for dissolved and particulate $^{231}$Pa and $^{230}$Th activities are the following (Rempfer

et al., 2017):

$$\frac{\partial A_d^j}{\partial t} = T(A_d^j) + \beta^j + K_{desorp}^j.A_p^j - (K_{adsorp}^j + \lambda_j).A_d^j \, , \tag{1}$$

$$\frac{\partial A_p^j}{\partial t} = T(A_p^j) - \frac{\partial(w_s.A_p^j)}{\partial z} - (K_{desorp}^j + \lambda_j).Ap^j + K_{adsorp}^j.A_d^j \, , \tag{2}$$

Where $A_d^j$ and $A_p^j$ are respectively the dissolved and particle-bound activities (dmp.m$^{-3}$.y$^{-1}$) of isotope j ($^{231}$Pa or $^{230}$Th), $\beta^j$ (dpm.m$^{-3}$.y$^{-1}$) is the water column production of isotope j by radioactive decay of its uranium parent, $\lambda_j$ (y$^{-1}$) is the decay constant

of isotope j, $w_s$ (m.y$^{-1}$) is the particle settling speed, $K_{adsorp}^j$ and $k_{desorp}^j$ (y$^{-1}$) are the adsorption and desorption coefficient of isotope j onto particles, respectively, T is the tracer balance evolution term (dpm.m$^{-3}$.y$^{-1}$) resulting from the water mass advection and diffusion terms computed by the CLIO ocean model. The values used in this study for each of the above-cited parameters are compiled in Table 1.

We chose to apply a uniform desorption coefficient denoted $K_{desorp}$ hereafter. However, the adsorption coefficient depends in

our model on the particle concentration and composition of each location and is calculated with the following equation (Rempfer et al., 2017):

$$K_{adsorp}^j(\theta, \Phi, z) = \sum_i \sigma_{i,j}.F_i(\theta, \Phi, z) \, , \tag{3}$$

Where $\theta$ is the latitude, $\Phi$ the longitude, z the water depth, $\sigma_{i,j}$ the scavenging efficiency for isotope j on particle type i (m$^2$.mol$^{-1}$), and $F_i$ is the particle flux (mol.m$^{-2}$.y$^{-1}$).

### 2.2 Model tuning and validation

The scavenging efficiency $\sigma_{i,j}$ is related to the partition coefficient $K_d$, which defines the proportion of each isotope ($^{231}$Pa or $^{230}$Th) lodged in the dissolved phase or bound to particles. Therefore one $K_d$ can be defined for each isotope and each particle type considered in the model ($K_{d(i,j)}$, j representing the isotope Pa or Th and i the particle type).

$$Kd_{(i,j)} = \frac{\sigma_{i,j} \times w_s \times \rho_{sw}}{M_{(i)} \times k_{desorp}} \, , \tag{4}$$

where $Kd_{(i,j)}$ is the partition coefficient for isotope i for particle type j, $\sigma_{i,j}$ is the corresponding scavenging efficiency, $w_s$ is the settling speed, $K_{desorp}$ is the desorption coefficient, $M_{(i)}$ is the molar mass of particle type i (i.e. 12 g.mol$^{-1}$ for POC, 100.08 g.mol$^{-1}$ for CaCO$_3$ and 67.3 g.mol$^{-1}$ for opal) and $\rho_{sw}$ is the mean density of sea water (constant fixed to 1.03 10$^6$ g.m$^{-3}$). Additionally the way each particle type fractionates the Pa and the Th is defined by the fractionation factor F(Th/Pa)$_j$.



$$F(Th/Pa)_i = \frac{Kd_{(Th,i)}}{Kd_{(Pa,i)}} = \frac{\sigma_{(Th,i)}}{\sigma_{(Pa,i)}},$$ (5)

$K_d$ and fractionation factors ($F(Th/Pa)$) have been measured for both radionuclides in various areas of the modern ocean and they show a rather large distribution (see Table S1) (Chase et al., 2002; Hayes et al., 2015). Consequently, these values are currently considered as tunable parameters in modelling studies (Dutay et al., 2009; Marchal et al., 2000; Siddall et al., 2005).

Considering three particle types for both radionuclides, there are thus six tunable $\sigma_{i,j}$ parameters in our model.

To efficiently sample our parameter space, we used a Latin Hypercube Sampling (LHS) methodology (https://CRAN.R-project.org/package=lhs). In order to only select parameters values consistent with observed $F(Th/Pa)$, we chose to use the couples $\{\sigma_{(Th,j)}, F(Th/Pa)_i\}$ as input parameters for the LHS. The value of $\sigma_{Pa,i}$ is then deduced from those two following equation (Eq. 5). The parameter ranges used in the LHS are given in Table S2. The LHS allowed a relatively good exploration of the

parameter space with a relatively small number of model evaluations. We performed 60 tuning simulations of 1,000 years each under pre-industrial boundary conditions. Consistently with previous modelling studies (van Hulten et al., 2018), this simulation length was sufficient for Pa and Th to reach an approximate steady state at the surface and in the deeper ocean. The model performance was evaluated by comparing outputs with present day particulate and dissolved water column Pa and Th measurements compiled in (Dutay et al., 2009) and references therein as well as sedimentary Pa/Th core tops data (Henderson

et al., 1999). We selected the ensemble member that best fits the observational constraints using 5 metrics corresponding to the Root Mean Square Error (RMSE) between observation data and the closest model grid cell average for particulate and dissolved Pa and Th as well as sedimentary Pa/Th (see text S1 for additional information). Table 2 presents the best-fit $\sigma_{ij}$ values. The best fit simulation is then used to investigate the multi-proxy response to abrupt circulation changes.

**2.3 Experimental design**

With the best fit $\sigma_{ij}$ values (see Table 2), we ran our model for 5,000 years under Pre Industrial (PI) conditions from a simulation with an equilibrated carbon cycle (Bouttes et al., 2015). The result of this equilibrium simulation is used as a starting point to perform hosing experiments of 1,200 years duration. The freshwater was added in the Nordic Seas following the approach described in (Roche et al., 2010). Each simulation contains three phases: a control phase (300 years), a hosing phase (300 years) and a recovery phase (600 years). The control phase is used to assess the natural variability of the circulation and associated

proxies under the PI climate state.

**3 Results**

**3.1. Model-data comparison under pre-industrial conditions**

The main goal of our study is to assess the response of two carbon-based proxies ($\delta^{13}C$, $\Delta^{14}C$) and of the Pa/Th to an abrupt circulation change in a physically consistent framework. This work represents a first step toward a better understanding of

marine multi-proxy records across the last glacial abrupt events. Therefore, our model-data comparison focuses on the Atlantic ocean, where the Pa/Th can be used as a kinematic circulation proxy (*e.g.* (François, 2007; McManus et al., 2004)). and on the simulated bottom particulate Pa and Th activities which can be directly compared to the Pa/Th ratio recorded in marine sediments. The water column concentration results are presented in the supplementary text S2.

The particle-bound Pa/Th of the deepest oceanic model grid cells is shown in Figure 1. The observations from core-top data are

superimposed as circles. Even if the observational dataset is somewhat patchy, it generally shows lower sedimentary Pa/Th ratios (0.04 or lower) in the basin interiors compared to the coastal areas. Another interesting feature is the high sedimentary Pa/Th values in the Southern Ocean between 50 and 75°S (opal belt), where Pa is heavily scavenged to the sediments by opal. Our model generates higher sedimentary Pa/Th in this region compared to the deep basin interior but our modelled opal belt stands slightly northward compared to the observations (Figure 1). In our best fit simulation, the adsorption/desorption




coefficients fall in the upper range of observations (see Table 2, Table S1), therefore, Pa and Th are very effectively scavenged to the sediments. Besides, while Pa is mostly scavenged by the opal particles, Th mainly reaches the sediments with $CaCO_3$ particles (see SOM for detailed information). Overall, these results are comparable with GCM outputs (van Hulten et al., 2018). The modelled values display the first order characteristics observed in the modern ocean and the sediment core tops. In the

following section, we test the sensitivity of the simulated sedimentary Pa/Th to abrupt circulation slowdown (hosing experiments).

**3.2 Multi proxy response to an abrupt circulation slowdown**

We added a freshwater flux of 0.3 Sv in the Nordic Seas during 300 years, which was sufficient to cause a drastic circulation reduction (Roche et al., 2010). Under PI conditions, the maximum of the AMOC stream function is about 15 Sv in our model.

During the hosing, North Atlantic water formation drops to nearly 0 Sv and the upper overturning cell completely vanishes (see Figure S3). The AMOC recovers ~200 y after the end of the water hosing and displays a small overshoot with the maximum Atlantic meridional stream function exceeding 20 Sv around 900 years.

We evaluate the response of Pa/Th, $\delta^{13}C$ and $\Delta^{14}C$ to the hosing in the Nordic Seas as follows. We identify for each model grid cell and each proxy, the simulation periods exceeding 80 years during which the proxy values are outside of their natural

variability range, defined as the proxy variance ($2\sigma$) under PI conditions over the first 300 years of the simulation (control phase-see Figure 2). In most cases, 0, 1 or 2 periods of significant response are detected. In some grid cells with high proxy variability, we detect up to 4 periods of significant proxy response, which are difficult to relate either with the hosing or overshoot timing of our simulation. Consequently, we excluded the grid cells depicting more than two periods of significant response from the subsequent analysis. For the time series containing two or less periods of interest, we call "time of maximum response", the

simulation year for which the absolute difference between the proxy value and mean proxy value during the control phase of the simulation is maximal. The proxy value at the time of maximum response is denoted "proxy response" in the following (Figure 2). The unique or dual proxy response is compared to the control proxy value (*i.e.* the mean proxy value over the 300 first years of the simulation). Figure 3 represents the zonal mean proxy response in the western Atlantic basin, the delimitation between the two basins being defined by the position of the mid-Atlantic Ridge (see Figure S4 and S5 for Eastern basin).

The $\delta^{13}C$ and $\Delta^{14}C$ only display one single response in the deep western Atlantic whereas the Pa/Th generally displays one or two responses in this part of the basin. Generally, the unique or the first response of each proxy has the same geographical pattern (Figure 3A. and B. early response) while in the case of two distinct responses, the late response has a radically different pattern (Figure 3 B. late response). For $\delta^{13}C$ and $\Delta^{14}C$, the late response corresponds to a general increase in the western Atlantic basin. For Pa/Th, the late response pattern consists in increased values in the southern Atlantic and decreased values in the

North-Atlantic and is the opposite of the early response pattern. For the three proxies, the late response pattern is generally consistent with the circulation overshoot observed around 900 simulated years.

Considering that the unique and the first responses represent the proxy response to the hosing, a consistent proxy response is observed in the following three zones of the western Atlantic basin: the surface and intermediate waters (0-1,500 m), the deep North-Atlantic waters (> 2,000 m), and the Southern Ocean (south of 30°S). In the surface waters and intermediate waters the

proxy response pattern is as follows. The $\delta^{13}C$ decreases in the first 500 m and then increases around 1,000 m. The $\Delta^{14}C$ increases from the surface to 1,500 m and Pa/Th displays no clear trend. In the deep North-Atlantic waters, the $\delta^{13}C$ and $\Delta^{14}C$ decrease while the Pa/Th generally increases, the three proxies reflecting a reduction of the North Atlantic Deep Water (NADW) flow rate. In terms of magnitude, we observe the strongest proxy response between 60°N and 40°N and between 1,500 and 3,000 m for the three proxies. In the Southern Ocean, the Pa/Th generally decreases, except in the deep southern basin (between 40 and

60°S below 3,000 m). The $\delta^{13}C$ and $\Delta^{14}C$ display a dipole pattern increasing at the surface and decreasing at depth. For $\delta^{13}C$, the depth boundary is around 1,500 m while for $\Delta^{14}C$ the depth boundary is ~1,500 m north of 50°S and reaches 3,000 m south of 40°S.



Looking at the proxy response times (Figure 4), we observe significantly different patterns for Pa/Th and the carbon isotopes. The $\delta^{13}$C and $\Delta^{14}$C response time increases with depth: in the surface and intermediate waters the response occurs roughly 300 years after the beginning of the fresh water addition, around 3,000 m the response is delayed by 150 years and towards the ocean bottom the delay increases up to 600 years. On the contrary, the Pa/Th displays much smaller response times, the timing of the Pa/Th response being generally synchronous with the minimum of the stream function 300 years after the beginning of the freshwater addition. Consequently, in most of the western Atlantic basin, the response of the carbon isotopes lags the Pa/Th response by a few hundred years, especially in the deeper waters.

In the eastern Atlantic basin the general pattern of the proxy response is similar to that of the western basin (Figure S4 - Figure S5). However, we note that the $\delta^{13}$C has frequently more than a single response, especially at depths > 2,000m. The $\delta^{13}$C displays overall the same pattern as in the western basin except in the south deep basin (between ~20 and 30 °S – below 2,500 m) in the case of a dual response. The $\Delta^{14}$C shows the same pattern as in the western basin, with increased values during the hosing in surface and intermediate waters and lower values at depth. Besides, the Pa/Th has a more complex pattern with a large increase in the arctic basin, a moderate increase in the tropical basin and a decrease in the northern and southern basins through the entire water column. Concerning the response times, the same pattern as in the western ocean is observed. Interestingly, we note that the $\Delta^{14}$C has longer response times in the eastern basin compared to the western basin in the case of one single response (Figure S5).

Overall, the objective analysis of the multi-proxy response allows the identification of 3 regions with different reaction patterns:
i) The clearest proxy response happens in the North western Atlantic between 40°N and 60°N, 1,000 and 5,000 m. In this area, the $\delta^{13}$C and $\Delta^{14}$C display marked decreases during the hosing while the Pa/Th significantly increases. Another interesting feature is that the $\delta^{13}$C and $\Delta^{14}$C lag the Pa/Th by about 200 y (Figure 5-A). In our model, this zone corresponds to the region of NADW formation and first portion of southward flow at depth.
ii) In the tropical intermediate and surface waters, $\delta^{13}$C and $\Delta^{14}$C increase during the hosing compared to the reference value while Pa/Th generally displays no clear reaction or very low amplitude increase (Figure 5-B).
iii) In the Southern Ocean (south of ~30°S), below 3,000 m, $\delta^{13}$C and $\Delta^{14}$C display a progressive decrease, while Pa/Th slightly increases during the hosing. No lag between the Pa/Th and the carbon isotopes response is observed (Figure 5-C). In our model, this zone corresponds to the Antarctic Bottom Water (AABW).

In short, in our model the three proxies considered do not systematically respond simultaneously and in a simple manner to a given oceanic circulation change. In the following section we discuss the underlying physical mechanisms behind these different proxy responses and we discuss how this result is informative for paleo-data multi-proxy studies that aim to reconstruct past circulation changes.

## 4 Discussion

### 4.1 Proxies and underlying physical mechanisms

An interesting feature of our simulations is that the $\delta^{13}$C and $\Delta^{14}$C response lags the Pa/Th response by a few hundred years, in particular in the Northwest Atlantic, bathed by the NADW. A potential explanation for this response time difference can be found in the physical and chemical mechanisms producing $\delta^{13}$C, $\Delta^{14}$C and Pa/Th signals. The balance between those mechanisms is fundamentally different for the carbon isotopes and the sedimentary Pa/Th. Indeed, on the one hand, the Pa/Th depends on the scavenging efficiency and on the water mass advection by the general circulation. The $^{231}$Pa and $^{230}$Th have relatively short residence time in the water column (50-200 years and 10-50 years respectively), and thus respond promptly to a circulation perturbation. Since we fixed the particles fields in our model, the scavenging efficiency for $^{231}$Pa and $^{230}$Th remains constant through the simulation. Consequently, our modelled Pa/Th response depends only on the $^{231}$Pa advection and thus, on the circulation changes. On the other hand, the oceanic carbon isotopes depend not only on circulation changes, but also on the



biological activity and the air-sea exchanges. Biological activity is perturbed by oceanic circulation changes as the latter modifies nutrient availability. Additionally, the different carbon reservoirs (terrestrial biosphere, ocean and atmosphere) exchange with each other. Consequently, the carbon isotopes need longer time to adjust to a circulation perturbation. This is consistent with the equilibration times that are required to equilibrate the proxies: 10, 000 years (Bouttes et al., 2015) for the

carbon cycle against 1,000 years for the Pa/Th (this study). In previous studies focusing on Pa/Th modelling, the equilibration time was even reduced to 500 years (van Hulten et al., 2018) and considered as a quasi-equilibrium. Thus, regarding the modelled processes on which the proxies rely, it appears logical that the Pa/Th has a shorter response time to a circulation change than the carbon isotopes.

### 4.2 Comparison to proxy data

### 4.2.1 Context overview

The best analog for our hosing experiments applied to a PI state would be the 8.2 ky cal BP event (Alley et al., 1997), attributed to the drainage of the glacial Lake Agassiz (Hoffman et al., 2012; Wiersma and Renssen, 2006). However, this event is of short duration (~300 years) and if some benthic $\delta^{13}$C data are available (*e.g.* (Kleiven et al., 2008)), to date there is no sufficiently well-resolved $\Delta^{14}$C and Pa/Th records covering this event.

The Heinrich events and DO-cycles are good candidates to compare with our hosing experiments because they are strongly related to freshwater fluxes in the North Atlantic (*e.g.* (Broecker et al., 1990; Hemming, 2004)) and associated with circulation changes that have been documented in $\delta^{13}$C, $\Delta^{14}$C and Pa/Th records (Böhm et al., 2015; Henry et al., 2016; Lynch-Stieglitz, 2017; Waelbroeck et al., 2018). However, the available paleoproxy data is relatively sparse and the extensive time series are rare. In addition, to date there is no published combined Pa/Th, benthic $\delta^{13}$C and $\Delta^{14}$C record available for the same sediment

core. Nevertheless, remarkably comprehensive multiproxy records are available for the Iberian margin (Gherardi et al., 2005; Skinner and Shackleton, 2004), the Brazilian margin (Burckel et al., 2015, 2016; Mulitza et al., 2017; Waelbroeck et al., 2018) and the Bermuda Rise (Böhm et al., 2015; Henry et al., 2016; Lippold et al., 2009; McManus et al., 2004). A classical working hypothesis is to assume that these records represent the proxy evolution of the surrounding basin. Besides, some recent compilations (*e.g.* (Lynch-Stieglitz et al., 2014; Ng et al., 2018; Zhao et al., 2018)) bring some insight about the evolution of

Pa/Th, benthic $\delta^{13}$C and $\Delta^{14}$C across the last 40 ky in the Atlantic Ocean.

However, the millennial scale climate changes of the last glacial cycle are not direct analogues of our hosing experiments because i) they occurred under glacial conditions whereas our simulations were run under PI conditions and ii) the Heinrich and DO events have distinct proxy patterns and cannot be entirely explained by a simple fresh water addition in the North Atlantic. Furthermore, the sequence of mechanisms involved in Heinrich stadials is still under debate (Barker et al., 2015; Broecker,

1994) and these periods were likely subdivided in several distinct phases (Ng et al., 2018; Stanford et al., 2011).

The model set up used in this study is rather crude. It uses fixed particle fields simulated by NEMO-PISCES that are overall satisfactory even if some unavoidable discrepancies with the observations have been pointed out. Among the most important features, the modelled concentrations are underestimated for POC and CaCO3, whereas they are overestimated for opal, especially along the western margins (van Hulten et al., 2018). It is therefore expected that deficiencies are apparent in the

simulated sedimentary Pa/Th in iLOVECLIM at those locations. Also, to date the iLOVECLIM Pa/Th module does not exhaustively include the processes governing Pa and Th scavenging. For instance, the preferential scavenging of Pa at continental margins (boundary scavenging) or by resuspended particles at the bottom in nepheloid layers (*e.g.* (Anderson et al., 1983; Thomas et al., 2006)) have not been parametrized in our model. This could explain the relatively high sedimentary Pa/Th values in the Northwest Atlantic offshore Newfoundland and Florida where 1) the CaCO$_3$ concentrations are known to be

underestimated and 2) there is evidence for important nepheloid layers (Gardner et al., 2018). Thereby, our model set up is mostly valid for sensitivity experiments and to investigate about the timing of the proxy responses. Thus, in the following



paragraphs we will only discuss the common/divergent trends rather than precise values between the paleodata and our model output.

### 4.2.2 General Atlantic features

The available paleodata display a consistent proxy evolution in the western Atlantic and in particular in the western boundary
current. Deep Northwest Atlantic cores present the same Pa/Th pattern over the last 25 ky with a significant Pa/Th increase during HS1 (Ng et al., 2018). A ~0.4 ‰ benthic $\delta^{13}C$ decrease is observed for some Heinrich events in deep and intermediate Atlantic waters (Lynch-Stieglitz et al., 2014). Finally, several coral and foraminifers $^{14}C$ records also depict a decreased $^{14}C$ (decreased $\Delta^{14}C$) content of deep waters and increased deep ventilation ages during HS1 (Chen et al., 2015; Skinner et al., 2014). While no major East-West difference has been highlighted concerning the carbon isotopes, Pa/Th response seems less
clear in the eastern Atlantic basin compared to the western boundary current. If some East Atlantic cores located close to the Mid-Atlantic ridge and the Iberian margin reproduce the amplitude of Pa/Th variations observed in the western basin, two other eastern records display no millennial scale variability (Ng et al., 2018).

In line with the paleoproxy data, our model results show a coherent and significant $\delta^{13}C$ and $\Delta^{14}C$ decrease in the deep western Atlantic. Overall the same response pattern is obtained in our model in the eastern Atlantic basin with the exception of the deep
South East Atlantic (between ~20 and 30 °S – below 2500 m). In this region, we observe a slight increase of the $\delta^{13}C$ during the early response in the case of a dual response while the $\Delta^{14}C$ decreases during the single or early response. We note that the amplitude of this early $\delta^{13}C$ increase is rather small compared to the late $\delta^{13}C$ decrease. Moreover, in this region, the response time is rather short (~100 – 150 y). We argue here that in this particular region the $\delta^{13}C$ displays a two-phase response, the actual long-term response corresponding to the late response.

In the surface and intermediate waters, the modelled $\delta^{13}C$ decreases above 500 m and then increases around 1,000 m while $\Delta^{14}C$ generally increases from the surface to 1,500 m (Figure 3). This $\delta^{13}C$ pattern is consistent with changes in productivity reported in previous hosing experiments. Indeed, due to changes in winds and upwelling intensity and subsequent changes in nutrient availability, the productivity generally decreases in the North and equatorial Atlantic (Menviel et al., 2008). Therefore, the $\delta^{13}C$ of the photic zone decreases and the $\delta^{13}C$ of the remineralization zone (~1,000 m) increases as the export from $^{13}C$ depleted
carbon is reduced during the hosing compared to the PI equilibrium state. As reported in other model studies, the increased $\Delta^{14}C$ in the upper ocean reflects the accumulation of radiocarbon in the atmosphere and the upper ocean in the absence of carbon entrainment to deep water through North Atlantic deep-water formation (Butzin et al., 2005). The generally good ventilation state of the upper ocean indicated by both $\delta^{13}C$ and $\Delta^{14}C$ may also be related to an increase of the of vertical mixing (and ventilation) in the intermediate waters in the absence of North Atlantic deep convection cell (Schmittner, 2007). The pattern we
obtain in the intermediate waters (Figure 3 and 5 B) is consistent with a $\delta^{13}C$ dataset on the Brazilian margin (Lund et al., 2015; Tessin and Lund, 2013) showing increasing $\delta^{13}C$ between 1,000 and 1,300 m and decreasing between 1,600 and 2,100 m water depth. These results are also fairly consistent with the modelled $\delta^{13}C$ described in (Schmittner and Lund, 2015), though our hosing experiments are fairly shorter (300 years vs 2,000 y).

In agreement with the Pa/Th data, the model simulates a clear Pa/Th increase in the NADW in the western basin while the
results are more ambiguous in the eastern basin. Indeed, in the eastern basin, except at high northern latitudes, very small amplitude Pa/Th variations are observed, with increasing values in the equatorial zone and decreasing values elsewhere. Thus, our modelled Pa/Th response at the Iberian margin is very different from the Iberian margin record. The low amplitude of our Pa/Th response may arise from the absence of strong Pa advection towards the Southern Atlantic in relation with the East Equatorial Atlantic upwellings variations across the hosing experiment (Menviel et al., 2008). Alternatively, the absence of
Pa/Th signal in the eastern basin could be due to an overestimation of the water fluxes that cross Gibraltar in the model. Indeed, the model resolution is rather coarse and the Gibraltar Straight is represented by a full ocean grid cell of 3 x 3 degrees, unrealistically impacting the simulated eastern Atlantic circulation. Additionally, the modelled Pa/Th displays interesting




features in the Southern Atlantic and in the surface and intermediate waters (between 40°S and 40°N). In both cases, no reliable Pa/Th record is available because particles fluxes from the opal belt (southern Atlantic) and coastal areas (surface waters) overprint the Pa/Th circulation signal. In our model, we observe that the single and the early responses do not match, and that the response of the Pa/Th is rather fast (~100 -150 y) in the case of a dual response. This suggests that in this area, the Pa/Th
displays a two-phase response to the freshwater addition and AMOC perturbation.

Further investigations about this two-phase response for Pa/Th and $\delta^{13}$C, in particular in the Southern Ocean, would require 1) monitoring of the detailed 3-D circulation pathway and carbon exchanges between the different reservoirs and 2) running experiments with increased hosing duration in order to better assess the proxy response to a sustained (~1,500 y) AMOC shutdown.

### 4.3.2. Bermuda Rise and Brazilian margin multi-proxy records

The Brazilian margin and the Bermuda Rise are located in the western boundary current and display similar proxy patterns. The Pa/Th increases from ~0.06 to the production ratio (0.093) or even above while the $\delta^{13}$C decreases by ~0.5 ‰ across the Heinrich events. Similar Pa/Th and benthic $\delta^{13}$C changes, but of smaller/reduced amplitude, are also observed for the DO stadials.

Below, we examine our modelled time series in the Bermuda Rise basin (~34°N-58°W, >4,300m) (Figure 6). The presented
time series correspond to the average of 9 model grid cells and are representative of the time series of the deep western basin (see Figure 5 and Figure 6). The simulated Pa/Th significantly increases between year 350 and year 850 of the simulation, consistently with the decrease of the maximum Atlantic stream function (Figure 6). The simulated Pa/Th approaches the production ratio of 0.093 at year 600, and the maximum stream function is close to zero from y 400 to 600, which is consistent with a sedimentary Pa/Th reaching the production ratio in the case of an AMOC shutdown. The simulated Pa/Th change has a
moderate amplitude of 0.015 Pa/Th units (from ~0.075 to 0.090) but is significant with respect to the natural variability recorded during the 300 first years of the simulation under PI conditions. Our simulated $\delta^{13}$C decreases from ~0.62 ‰ to ~0.5 ‰, reaching the minimum around year 900, i.e. 600 years after the beginning of the hosing. The simulated $\delta^{13}$C decrease has a moderate amplitude of ~0.12 ‰ but is significant with respect to the natural variability recorded in the control first 300 years of the simulation. Although the trend of the simulated and observed proxy response is the same, their absolute values differ. In our
simulated record, the Pa/Th value associated with the modern circulation scheme is around 0.075, which is significantly higher than the value actually measured at the Bermuda Rise or Brazilian margin sites: ~0.06. For both proxies, the simulated amplitude of change is much smaller than the amplitude recorded in the paleodata across the Heinrich events: the modelled $\delta^{13}$C decrease is around 0.12 ‰ while it is 0.5 ‰ in the paleodata; the modelled Pa/Th change is ~0.015 while it is ~0.03 in the paleodata. This might be a consequence of the short duration of the fresh water forcing and induced reduction in AMOC (for carbon
isotopes) and to poor particle representation along the Northeast American coast (for Pa/Th).

The lead/lag relationship between Pa/Th and benthic $\delta^{13}$C was previously examined on data both at the Brazilian margin and the Bermuda Rise with opposite conclusions: at the Brazilian margin the Pa/Th was found to lead $\delta^{13}$C by about 200 years (Waelbroeck et al., 2018), while at the Bermuda Rise, the Pa/Th was found to lag $\delta^{13}$C by about 200 years (Henry et al., 2016). However, the latter result must be interpreted with caution since the cross-correlation method used to evaluate the lead or lag
relationships in (Henry et al., 2016) has been designed for signals that are stationary in time and is thus is not suitable to analyze non-stationary climatic signals (Waelbroeck et al., 2018). Therefore, the paleodata seem to suggest a lead of the Pa/Th response with respect to the carbon isotopes response in the western boundary current. However, this observation may simply reflect the impact of bioturbation on sediment archives. Indeed, even if the two proxies are recorded in a single sediment core, it is important to note that both proxies are not hosted by the same sediment fraction: the Pa and Th being preferentially adsorbed
on the fine grain particles (<100 µm, (Chase et al., 2002; Kretschmer et al., 2010; Thomson et al., 1993)) while $\delta^{13}$C and $\Delta^{14}$C are measured on foraminifer shells that correspond on larger particle sizes (> 150 µm). It has been shown that bioturbation could affect different particle sizes differently (*e.g.* (Wheatcroft, 1992)). Therefore Pa/Th and carbon isotopes could be affected by



bioturbation in a different way, and the 200 years Pa/Th response lead on carbon isotopes observed in Brazilian margin sediments could then solely be explained by sediment bioturbation, as suggested in (Waelbroeck et al., 2018). In contrast, in our model we observe a systematic lead of the Pa/Th response with respect to the carbon isotopes in the NADW. Therefore, our model suggests that this Pa/Th lead may be a feature of the proxy response to millennial scale variability and is not necessarily

an artefact due to the marine core bioturbation.

**Conclusions and perspectives**

We have implemented the $^{231}$Pa and $^{230}$Th tracers in the climate model of intermediate complexity iLOVECLIM. The new Pa/Th module simulates dissolved and particulate $^{231}$Pa and $^{230}$Th concentrations with a quality comparable to GCMs under PI equilibrium. The model represents well the open ocean sedimentary Pa/Th ratio. The largest model-data discrepancies are

observed for coastal areas and arise from 1) the fixed particle fluxes used in our model set-up and 2) the lack of representation of processes affecting the $^{231}$Pa and $^{230}$Th such as the nepheloid layers or sediment resuspension at the ocean bottom.

To date, the model is able to simulate the evolution of $\delta^{13}$C, $\Delta^{14}$C and Pa/Th over thousands of years in a consistent physical framework and in a reasonable computation time (~800 years per 24h). We tested and fingerprinted the response of these three proxies to an imposed and abrupt circulation change by performing hosing experiments. We analysed the results corresponding

to a significant circulation reduction, corresponding to a freshwater input of 0.3 Sv for 300 years in the Nordic Seas. For the three proxies, we detect between 0 and 4 periods during which the proxy value significantly differs from its PI value. As results with more than 2 periods of significant change are delicate to interpret in terms of proxy response to the freshwater forcing, we focused the study on the analysis of the time series exhibiting 0, 1 or 2 distinct periods of significant change. Based on the anomaly between each proxy value during the hosing and the reference values under PI conditions, we highlight 3 distinct

patterns in the proxy responses, which correspond to the three main water masses of the Atlantic Ocean: NADW, AAIW and AABW. In the intermediate waters, the Pa/Th and $\Delta^{14}$C show a slight increase while the $\delta^{13}$C significantly increases. In the AABW, the three proxies display consistent and synchronous response: the Pa/Th increases while the $\delta^{13}$C and $\Delta^{14}$C decrease, corresponding to a reduced circulation and ventilation of the water mass. Finally, in the NADW, the three proxies display consistent responses too, with the Pa/Th increasing and the $\delta^{13}$C and $\Delta^{14}$C decreasing. The response time of $\delta^{13}$C and $\Delta^{14}$C is

generally similar and increases with increasing water depth, while Pa/Th displays a more homogeneous and shorter response time. The maximum response of the $\delta^{13}$C and $\Delta^{14}$C is simulated in the NADW and displays a lag of about 250 years with respect to the Pa/Th response.

We argue that this lag between the carbon isotopes and Pa/Th responses can be explained by the fundamentally different mechanisms at play to produce the proxy record. Indeed, in our model, because our scavenging intensity is kept constant due to

the fixed particle fields, the Pa/Th only depends on circulation. Besides, the Pa and Th have relatively short residence times in the ocean water column, making the Pa/Th proxy prompt to respond to any circulation change. On the contrary, the carbon isotopes in the ocean interact with the carbon from other reservoirs, via air-sea exchanges and biological activity, which takes time.

Even if our model results mainly consist in sensitivity tests and are not suitable to extensive model-data comparison, we observe

some features consistent with the available paleo proxy record: we observe i) coherent proxy response along the Atlantic western boundary current for the three proxies corresponding to a significant Pa/Th increase and $\delta^{13}$C and $\Delta^{14}$C decrease and ii) distinct proxy responses for intermediate and deep waters for $\delta^{13}$C and $\Delta^{14}$C. Besides, our model experiment suggests that there is a constitutive lag of the carbon isotopes over the Pa/Th response to a freshwater input in the NADW water mass. Such a lag has been evidenced in one proxy record at the Brazilian margin although it was suspected to result from attributed to sediment

bioturbation bias as the two proxies are carried by particles of different sizes.





The observed discrepancies between our model experiment and the proxy data in this study can be partly attributed to the incomplete process representation in our model. In particular, because we have fixed particles in our model we cannot capture the totality of the Pa/Th signal. Future work would require the evaluation of the multi-proxy response in a more realistic numerical experiment, using glacial boundary conditions and coupled/interactive particle fields. In addition, a more complete

5   dataset containing multi-proxy records is needed to achieve a more complete model-data comparison.



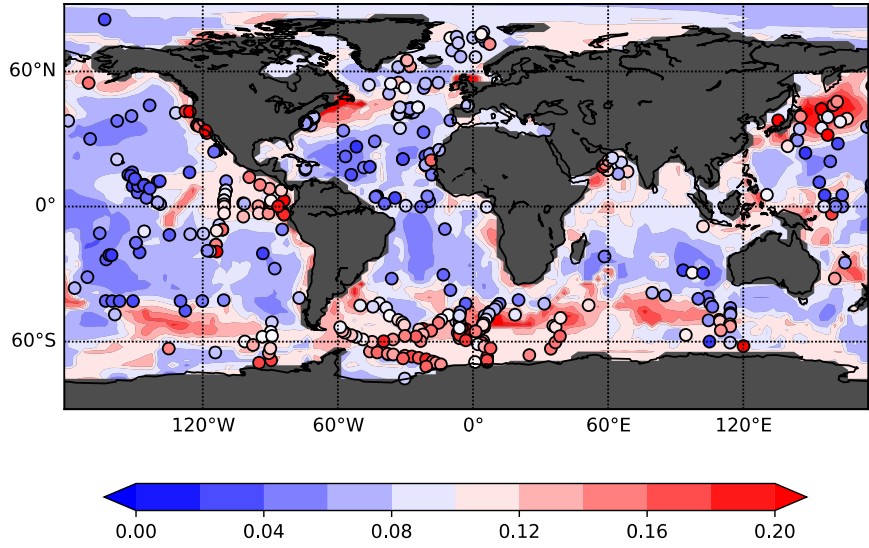

**Figure 1:** Map showing the particulate $^{231}$Pa/$^{230}$Th activity ratio of the deepest model ocean grid cells. The simulated Pa/Th ratio is represented in the colour background. The observations compiled in (Henderson et al., 1999) are represented as circles.





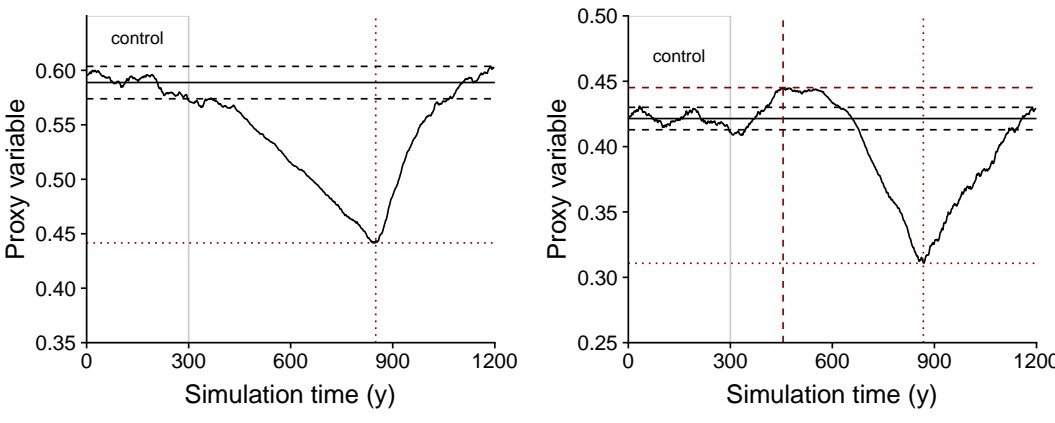

**Figure 2**: Definition of the proxy response (dotted red vertical line) and the response time (dotted red horizontal line) in the case of A. a single response or B. a dual response to the circulation change induced by the freshwater addition. In the case of dual response, we examine the early (*i.e.* the first) and the late (*i.e.* the second) proxy response and response time. The anomaly is defined as the difference between the proxy value at the defined response time and the average proxy value during the control period of 300 years (highlighted in grey).





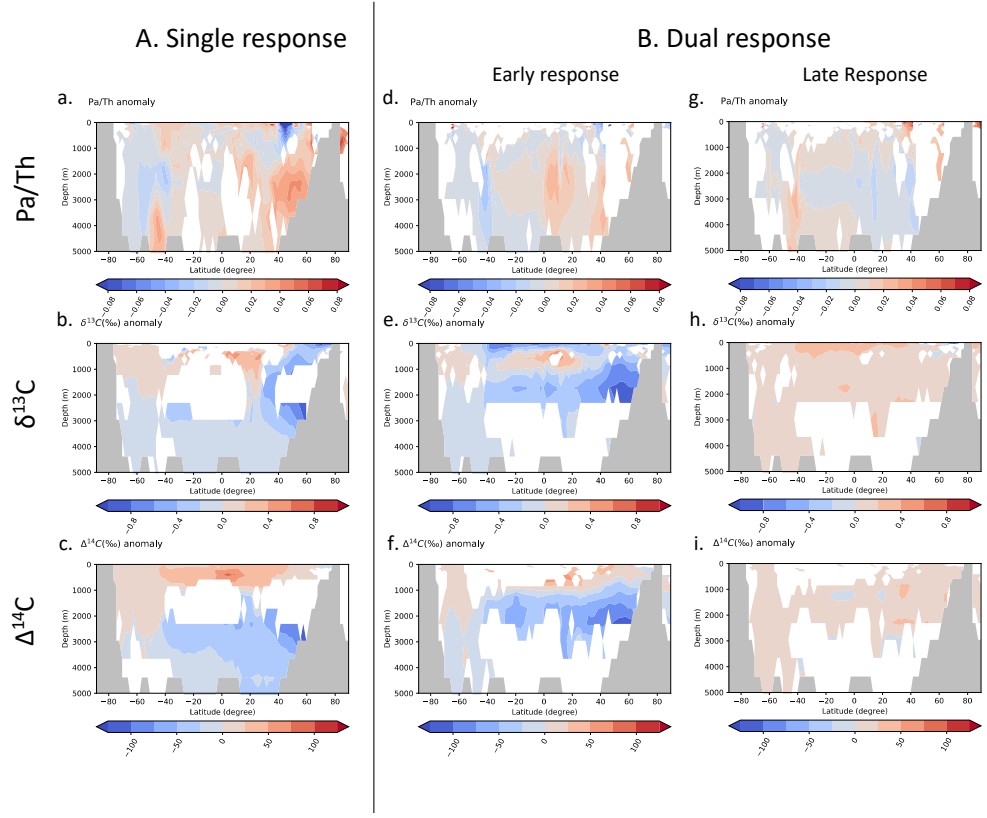

**Figure 3: Zonally averaged anomalies for Pa/Th, δ¹³C and Δ¹⁴C in the western basin in the case of a freshwater input in the Nordic seas.** The western and eastern basin are delimited by the topography increase corresponding to the Mid Atlantic Ridge in the model grid. The anomalies are computed/defined as the proxy response minus the mean of the proxy value during the control period of 300 years under PI conditions (see text). **A.** (a. to c.) Represents the anomalies for the three proxies in the case where exactly one proxy response has been detected. In the case of two proxy responses, (**B.**) d.to f. represent the proxy anomaly value for the early (first) response, while g.to i. represent the proxy anomaly for the late (second) proxy response. Areas left in blank were not showing a unique response (A.) or not showing exactly two responses (B.) In each subplot, the grey contours represent the ocean bottom.





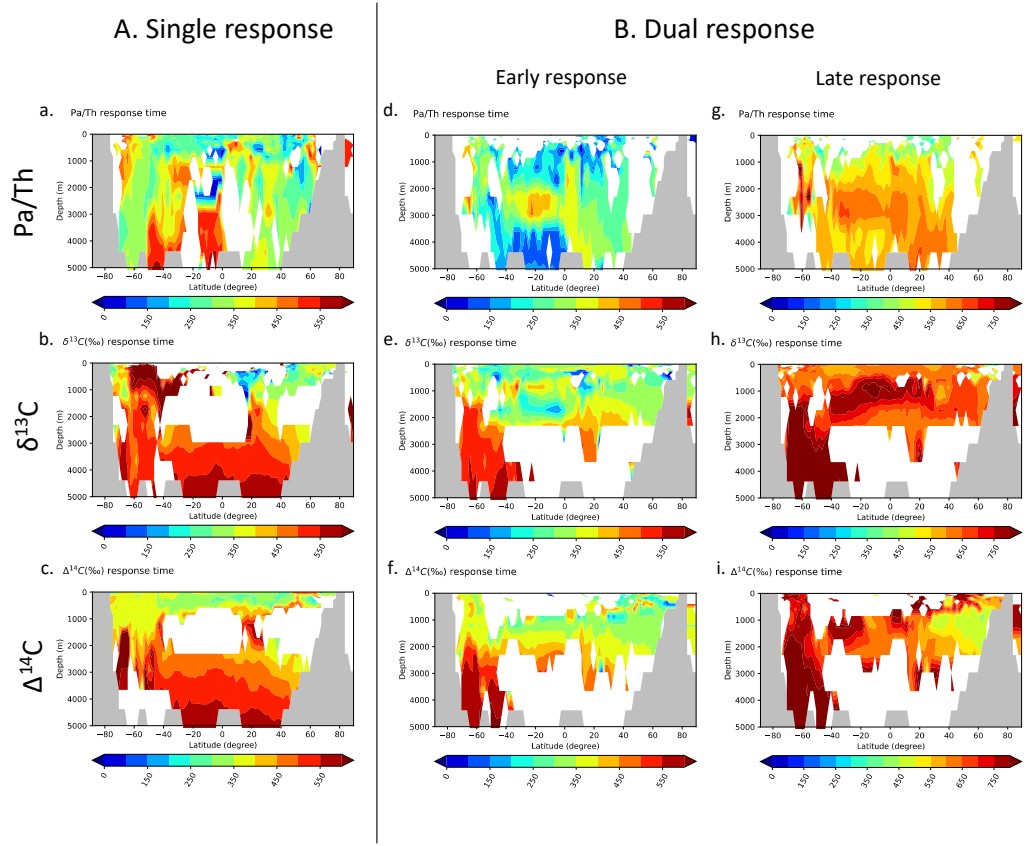

**Figure 4: Zonally averaged times of response for Pa/Th, δ¹³C and Δ¹⁴C in the western basin in the case of a freshwater input in the Nordic seas.** The times of response correspond to the time of proxy maximal response (see text). **A.** Response time in the case where exactly one single proxy response is detected. In the case where two distinct responses are detected **(B.)** d. to f. show the response time of the early (first) response and g. to i. show the response time corresponding to the late (second) response. Areas left in blank were not showing a unique response (A.) or not showing exactly two responses (B.) In each subplot, the grey contours represent the ocean bottom.





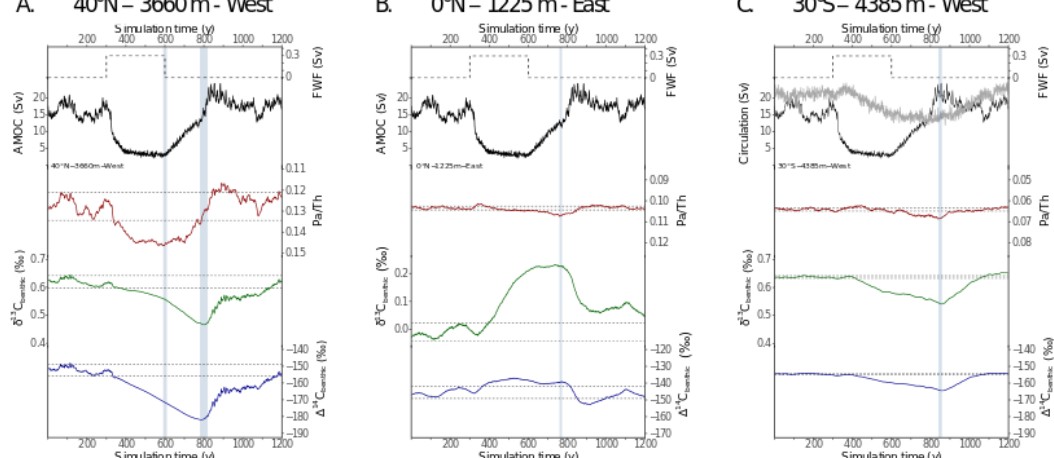

**Figure 5: Selected multi-proxy time series for the Nordic Seas hosing experiment representing the three main Atlantic water masses**.
**A.** Zonally averaged on the western Atlantic basin time series at 40°N 3660 m. This time series is representative of the proxy behaviors in the NADW. **B.** Zonally averaged on the eastern Atlantic basin time series at 0°N 1225 m. This time series is representative of the intermediate waters. **C.** Zonally averaged on the western Atlantic basin time series at 30°S 4385 m. This time series is representative of the proxy response in the AABW. In the eastern basin, the $\delta^{13}$C generally displays 2 responses, first increasing (not shown) and then decreasing (as shown). In each subplot from top to bottom the dashed black line represent the freshwater flux (FWF) applied in the Nordic Seas, the black line represents the North Atlantic Maximum stream function (A. B. C.), the grey line represents the Southern Ocean maximum stream function (C.), the red line represents the particulate Pa/Th, the green line represents the $\delta^{13}$C and the blue line represents the $\Delta^{14}$C. The thin dashed black lines represent the proxy variance (2 σ) on the first 300 simulated years. The blue vertical bands indicate the timing of the proxy responses.

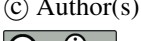


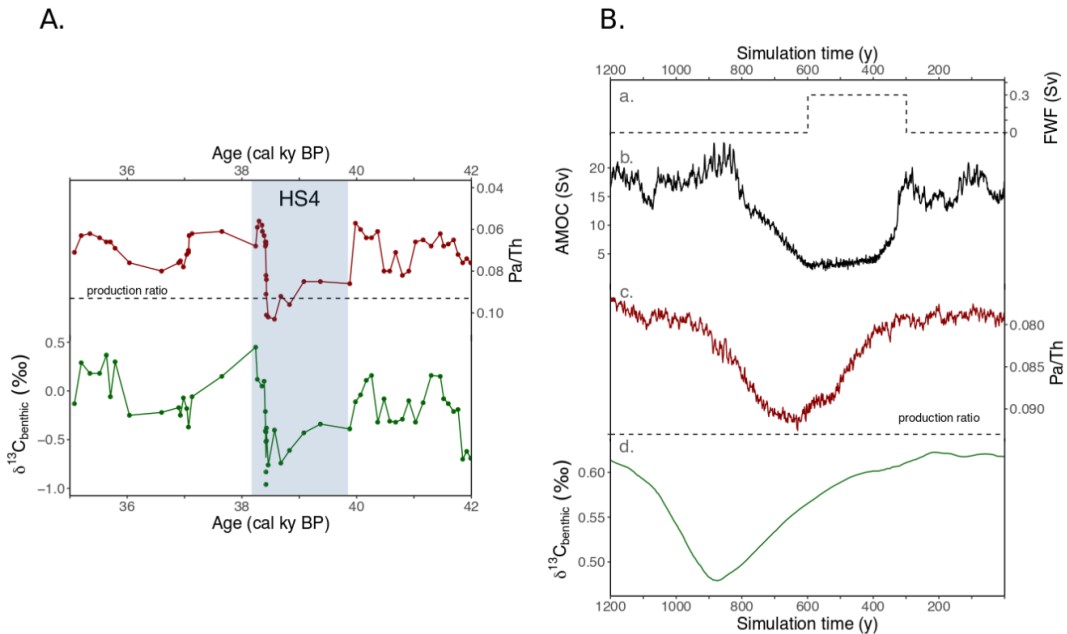

**Figure 6: Example of the Bermuda Rise time series A.** Pa/Th and benthic δ¹³C (‰) measured at the Bermuda Rise around HS4 (Henry et al., 2016). **B.** Modelled time series corresponding to the average of the 9 grid cells surrounding the model grid cell closest to the Bermuda Rise (34°N-58°W). a. Imposed freshwater flux, b. Maximum North-Atlantic meridional Stream function (Sv) c. simulated Pa/Th and, d. benthic δ¹³C (‰).



*Code and availability.* The iLOVECLIM source code is based on the LOVECLIM model version 1.2 whose code is accessible at http://www.elic.ucl.ac.be/modx/elic/index.php?id=289. The developments on the iLOVECLIM source code are hosted at https://forge.ipsl.jussieu.fr/ludus but are not publicly available due to copyright restrictions. Access can be granted on demand

5   by request to D. M. Roche (didier.roche@lsce.ipsl.fr)

The model output related to this article has been submitted to Pangaea (PDI-20335).

*Supplement.* The supplement related to this article is available online.

10   *Author contributions.* LM, CW and DMR designed the research. LM, DMR, NB, JCD and AQ developed the iLOVECLIM model. LM performed the simulations. LM and JYP developed the post-processing algorithm. JCD and SP contributed to expert knowledge on Pa/Th. LM wrote the manuscript with the inputs form all the co-authors.

*Competing interests.* The authors declare that they have no conflict of interests.

*Acknowledgements* This is a contribution to ERC project ACCLIMATE; the research leading to these results has received funding from the European Research Council under the European Union's Seventh Framework Programme (FP7/2007-2013)/ERC grant agreement 339108. LM acknowledges funding from the Australian Research Council grant DP180100048. We thank S. Moreira and F. Lhardy for their help with Python. E. Michel is thanked for expert knowledge discussion on [14]C.

20   This is a LSCE contribution 6601.



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

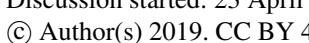



**Table 1: Parameters used in the Pa/Th module**

| Symbol | Variable | Value | Units |
|---|---|---|---|
| $j$ | $^{231}$Pa or $^{230}$Th | - | - |
| $i$ | Particle type (CaCO$_3$, POC, opal) | - | - |
| $A_p^j$ | Particle-bound activity | Calculated – see Eq. 2. | dpm.m$^{-3}$.y$^{-1}$ |
| $A_d^j$ | Dissolved activity | Calculated – see Eq. 1. | dpm.m$^{-3}$.y$^{-1}$ |
| $\beta^{Pa}$ | Production of $^{231}$Pa from U-decay | 2.33 10$^{-3}$ | dpm.m$^{-3}$.y$^{-1}$ |
| $\beta^{Th}$ | Production of $^{230}$Th from - decay | 2.52 10$^{-2}$ | dpm.m$^{-3}$.y$^{-1}$ |
| $\lambda_{Pa}$ | Decay constant for $^{231}$Pa | 2.116 10$^{-5}$ | y$^{-1}$ |
| $\lambda_{Th}$ | Decay constant for $^{230}$Th | 9.195 10$^{-6}$ | y$^{-1}$ |
| $K_{adsorp}^j$ | Adsorption coefficient | Calculated – see Eq. 3. | y$^{-1}$ |
| $k_{desorp}$ | Desorption coefficient | 2.4 | y$^{-1}$ |
| $\sigma_{i,j}$ | Scavenging efficiency | See Table 2 | m$^2$.mol$^{-1}$ |
| $F_i$ | Particle flux | Calculated in each grid cell | mol.m$^{-2}$.y$^{-1}$ |
| | | $Fi = $ [particle conc]. $w_s$ | |
| $w_s$ | Uniform settling speed | 1,000 | m.y$^{-1}$ |



**Table 2: Best fit $\sigma_{i,j}$ values and corresponding Kd values**

|  | $\sigma_{Pa\text{-}CaCO3}$ | $\sigma_{Pa\text{-}POC}$ | $\sigma_{Pa\text{-}opal}$ | $\sigma_{Th\text{-}CaCO3}$ | $\sigma_{Th\text{-}POC}$ | $\sigma_{Th\text{-}opal}$ |
|---|---|---|---|---|---|---|
| Best fit | 1.87 | 1.55 | 7.62 | 76.83 | 5.47 | 3.77 |
|  | $Kd_{Pa\text{-}CaCO3}$ | $Kd_{Pa\text{-}POC}$ | $Kd_{Pa\text{-}opal}$ | $Kd_{Th\text{-}CaCO3}$ | $Kd_{Th\text{-}POC}$ | $Kd_{Th\text{-}opal}$ |
| Best fit | 8.01E+06 | 5.53E+07 | 4.86E+07 | 3.29E+08 | 1.96E+08 | 2.40E+07 |