# Peer review of "Carbon isotopes and Pa/Th response to forced circulation changes: a model perspective"

_Climate of the Past, 2019_

## Referee Comment (RC1) · Anonymous Referee #1 · 8 Jul 2019

The authors implement Pa/Th in the intermediate complexity model LOVECLIM. With the carbon isotopes, which are already in the model, the authors evaluate the responses of different proxies to the freshwater fluxes in the North Atlantic in a classical hosing experiment. They find that the Pa/Th leads the carbon isotopes by a few hundred years in the deep Atlantic. Pa/Th has been implemented in different GCMS and the authors follow the approach in Rempfer et al. (2017). Also, modeled Pa/Th response to fresh water fluxes added to the North Atlantic is carried out in previous studies (Gu and Liu, 2017; Rempfer et al., 2017). However, the comparison between Pa/Th and carbon isotopes helps to distinguish this study with previous modeling studies. Revisions are needed before this could be acceptable for publication.

Major Comments: 1. I find the separation of the single response and the dual response

quite confusing. First, is it really to identify the responses this way? It seems that for the dual response, the first response is associated with the AMOC reduction and the second response is associated with the AMOC overshoot (as pointed out in page 6 line 30). I think it is easier for people to follow if you state this as a response to decreased AMOC or increased AMOC instead of first or late response. Secondly, why some grids (for example 40S, 4000m, Figure 3 a, d and g) show both single response and dual response?

2. Responses of Pa/Th and carbon isotopes in the Atlantic in a hosing experiment are not new and have been examined in other studies already. Since this paper focuses on Pa/Th, their modeled Pa/Th response should be compared to previous studies (Gu and Liu, 2017; Rempfer et al., 2017). Spatial and temporal similarities and differences with these previous studies should be compared and discussed.

3. At the end of the introduction, three questions are raised. The first two questions are discussed in section 3 and 4, but the third question "How can the modelled multi-proxy response help to interpret the paleoproxy records" is not clearly answered. The implication for interpreting the paleoproxy records is not clearly state. This is a very important question for modeling proxies in GGMS. Authors need to add some discussion about this kind of implications in the discussion.

4. More differences between Pa/Th and carbon isotopes in reconstructing past AMOC could be discussed and highlighted. As mentioned above, the novelty of this paper is studying the Pa/Th together with carbon isotopes since the Pa/Th and carbon isotopes in a hosing experiment have been presented in previous studies. However, I feel this multi-proxy comparison is not fully developed in the current manuscript. A more in-depth comparison between Pa/Th and carbon isotopes and their implications for paleoceanography (back to comment 3) are needed.

5. Pa/Th leads carbon isotopes, but lead by how many years? From Figure 5 a and c, it seems that the 300 years hosing is too short for carbon isotopes to fully adjust to

the reduction of AMOC. If hosing is kept longer than 300 years, carbon isotopes may lag Pa/Th response even longer. Therefore, from this 300-years hosing experiment, we cannot determine the exact lead time. This should be pointed out.

6. The modeled Pa/Th is compared to observations in Dutay et al., 2009 and Henderson et al., 1999 (Page 5, line 14). However, in recent years, many new observations are now available. GEOTRACES offers a lot of relevant new data (also used in Rempfer et al., 2017). More core top Pa/Th are also available. A more complete compilation of the observations should be used to tune the model parameters. Also, if comparing to the same compilation of observations as in previous studies (Gu and Liu, 2017; Rempfer et al., 2017; Van Hulten et al., 2018), model performance in simulating Pa/Th can be estimated quantitatively.

Minor Comments: 1. Page 3, Line 27, Gu et al. (2017) simulating Pa/Th in CESM should be mentioned here (higher resolution then Rempfer et al. (2017) and longer integration than van Hulten et al. (2018)). Gu, S., Liu, Z., 2017. 231Pa and 230Th in the ocean model of the Community Earth System Model (CESM1.3 ). Geosci. Model Dev. 10, 4723–4742. https://doi.org/10.5194/gmd-10-4723-2017

2. Page 4, Authors follow Rempfer et al. (2017) to implement Pa/Th. One advance in Rempfer et al. (2017) in simulating Pa/Th is that bottom scavenging and boundary scavenging are included, which improves the simulation of water column Pa and Th activity. In page 8, line 37, authors state that the bottom and boundary scavenging are not modeled in LOVECLIM. This should be mentioned earlier in section 2.1 (model description and developments). Also, the modeling scheme (similarities and differences) comparing with previous modeling efforts should be discussed explicitly in section 2.1.

3. Page 5, Line 20 Details about the PI forcing should be provided. From Figure 5, there is interannual variability. Is the PI forcing looping in the first 300 years?

4. Section 3.1 Vertical structures of Pa/Th could also be provided and compared to observations (GEOTRACES transects), such as Figure 2 and Figure 3 in Rempfer et

al. 2017. Figure S1 only have particulate and dissolved Pa and Th.

5. Page 6, section 3.2, first paragraph, Figure 5 can be referred here. Then people can see exactly how the fresh water is added and how the AMOC evolves.

6. Page 6, line 14-16, this sentence can be rewritten for easier understanding.

7. Page 7, line 15, Any explanations for the 14C response time difference between the eastern and western basin?

8. Figure 2 gives two examples of the single response and dual response. What is the proxy exactly? Pa/Th? 13C? or 14C? And where is the grid, location and depth? Also, it would be good to add AMOC in this plot for people to follow.

9. Authors use fixed particle fluxes in their hosing experiment. After adding fresh water to the North Atlantic, the particle fluxes will change. Will this particle flux change affect the results of this paper should be discussed.

10. The conclusions and perspectives can be improved to highlight the major findings. Currently, it is too broad and descriptive.
* * *

---

## Referee Comment (RC2) · Anonymous Referee #2 · 14 Sep 2019

This is a report on the implementation of the Pa/Th sedimentary proxy in the ocean-climate model of intermediate complexity, iLOVECLIM, in addition to the previously included stable carbon and radiocarbon isotope ratios. The reconstruction of past circulation states has suggested substantial changes from that observed in the modern ocean, with potentially significant implications for past climate change. It is therefore important that model simulations can capture the observed sedimentary evidence and demonstrate the ocean physics that might be consistent with this evidence. In this case, the incorporation of multiple isotopic tracers with different distribution and influences adds a valuable layer of sophistication to such modeling efforts.

In addition to demonstrating the model's ability to reproduce the observed modern distributions of Pa/Th and carbon isotopes, the authors report on the results of what is

now a relatively standard "hosing" experiment, wherein freshwater is imposed on the surface of the high latitude North Atlantic within the model domain, in order to weaken convection and the overturning circulation. Changes in subsurface water masses and the strength of the overturning have the result of redistributing the sedimentary Pa/Th and carbon isotopes, which the authors then interpret and compare to existing data. They identify different responses of in the respective tracers. One major finding is that in the hosing experiment, changes in both carbon isotopes lag Pa/Th by a few hundred years.

Overall, this study is an important step forward in terms of the state of the art of implementation of circulation proxies and should therefore be worth accepting for publication in Climate of the Past, following revisions that should address the following points.

A more careful data-model comparison is needed to validate the simulated Pa and Th, which is the main advance made in the model. The paper compares bottom water particulate Pa/Th with a core top compilation (Henderson et al., 1999). Other than that, the comparison with Pa/Th data is mostly qualitative. The authors acknowledge that they refrained from making more data/model comparisons because of the crudeness of the model (Page 8 Line 41 (P8L41)). However, it is still important to show those comparisons. Readers may gain information about the fidelity (or lack thereof) of the model to the modern observations, including regions where the model performs well and regions where it does not. This information will help make the audience more informed, and therefore increase the impact of the paper. For example, the paper mentions the compilations of sedimentary Pa/Th by Lippold et al. (2016) and Ng et al. (2018) (P9L6). How does the model compare with them graphically? How do the particulate and dissolved Pa and Th results compare with GEOTRACES observations? These could be addressed in a few brief passages.

Additionally, hypotheses are offered for why $\delta$13C response leads Pa/Th (possibly biology and/or air-sea exchange slows down $\delta$13C response), yet given the setup of the model, it would be a missed opportunity not to conduct a more detailed diagnosis of

modeled causes for the lead-lag relationships among the various tracers. If the reasons can be pinned down, the paper can make a more robust conclusion, even if it is model-depending. Is it possible to plot the biological changes before and after a hosing experiment? How about changes in the air-sea exchange rate? Depending on the results of those plots, the paper can then present a fuller picture of the changes during a hosing experiment.

Additional smaller points for consideration:

In P1L25 This is confusing. Should it read "without an a priori guess"?

In P1L33-34 These are not global changes and should be more narrowly defined, possibly as regional or even local.

In P1L36, it should be "see Lynch-Stieglitz (2017) for a review".

In P1L40, $\varepsilon$Nd, Cd/Ca, sortable silt are also valuable proxies to reconstruct circulation and water mass and they are worth mentioning.

In P2L25, this should be 12C, although in truth it is both, with a lower 13C/13C.

In P4L17, Equation 2, the second minus sign is different from the first. The multiplication dot is positioned as a punctuation would.

In P4L34, Equation 4, the "d" in "Kd" should be subscript.

In P5L14, it should be "compiled in Dutay et al. (2009)".

In P6L13, this is a long and potentially confusing sentence, yet a valuable one for its description of how the proxies were evaluated. It would help to have a comma after "identify", which might make it clearer that the identification is of simulation periods exceeding a defined length, for each proxy.

In P8L3-5, this is a bold statement that just does not ring true. Ten thousand years for equilibration of the carbon isotope signal in the Atlantic ocean, and a thousand for

Pa/Th? This is in a basin where the residence time of the deep waters is a few hundred years today, and maybe a thousand years more in the past. Unless I misunderstand the point here, something is not right.

In P8L26, when listing the reasons that the millennial scale climate changes are not analogous to the hosing experiments, it would be useful to also point out that the location of the freshwater hosing in the model (the Nordic Seas) could also be different from events that originate primarily from the Laurentide ice sheet on North American, mostly likely including the millennial Dansgaard-Oeschger and Heinrich events.

P9L10, substitute "If" with "While".

P9L32, modeled $\delta13C$ results are compared with another model's results, yet this section is named "Comparison to proxy data". Maybe update section title to "Comparison to proxy and modeled data."

In Figure 2, the labels (e.g. A. Single response) are unnecessarily far from the plots. The caption should define the dotted black lines (which I assume is the 2 sigma variation of the control phase). Also in the caption, the dotted red vertical line is the response time and the dotted red horizontal line is the proxy response. The caption states it the other way.

In Figure 3, I think I'm missing something here. Why are there overlaps between the data coverage of single and dual response plots? Shouldn't the two be mutually exclusive?

Lastly, a citation in your references has the wrong publication year. The citation for "Luo, Y., Francois, R. and Allen, S. E.: Sediment 231Pa/230Th as a recorder of the rate of the Atlantic meridional overturning circulation: insights from a 2-D model., Ocean Science Discussions, 6(4), 2755–2829, 2009." should instead be Luo, Y., Francois, R., and Allen, S. E.: Sediment 231Pa/230Th as a recorder of the rate of the Atlantic meridional overturning circulation: insights from a 2-D model, Ocean Sci., 6, 381-400,

https://doi.org/10.5194/os-6-381-2010, 2010.

---

## Author Comment (AC1) · 17 Dec 2019

**Carbon isotopes and Pa/Th response to forced circulation changes: a model perspective**

**Note to the Editor and referees**

We have requested a long deadline extension since a bug was found in the iLOVECLIM model code that could affect the carbon cycle part. To ensure scientific reproducibility, we first wanted to assess whether the bug found could have a significant impact on our results. This has required to run several multi-millennial long simulations. After analysis of the results of the newer version, we however found that our conclusions are unaffected by this error. We thank the Editor and the referees for their patience in this necessary process.

**Response to the referees' comments**

We thank both reviewers for their constructive comments that helped to improve and clarify the manuscript. We have addressed the comments in detail below.

**Anonymous Referee #1**

The authors implement Pa/Th in the intermediate complexity model LOVECLIM. With the carbon isotopes, which are already in the model, the authors evaluate the responses of different proxies to the freshwater fluxes in the North Atlantic in a classical hosing experiment. They find that the Pa/Th leads the carbon isotopes by a few hundred years in the deep Atlantic. Pa/Th has been implemented in different GCMS and the authors follow the approach in Rempfer et al. (2017). Also, modeled Pa/Th response to fresh water fluxes added to the North Atlantic is carried out in previous studies (Gu and Liu, 2017; Rempfer et al., 2017). However, the comparison between Pa/Th and carbon isotopes helps to distinguish this study with previous modeling studies. Revisions are needed before this could be acceptable for publication.

Major Comments:

1. I find the separation of the single response and the dual response quite confusing. First, is it really to identify the responses this way? It seems that for the dual response, the first response is associated with the AMOC reduction and the second response is associated with the AMOC overshoot (as pointed out in page 6 line 30). I think it is easier for people to follow if you state this as a response to decreased AMOC or increased AMOC instead of first or late response. Secondly, why some grids (for example 40S, 4000m, Figure 3 a, d and g) show both single response and dual response?

We thank the reviewer for highlighting the importance of the chosen terminology. We agree with the reviewer, in numerous locations, the "first response" seems to be associated with the AMOC reduction and the "the second response" with the AMOC

overshoot. However, this is **not always true**. For instance, as pointed out in the manuscript, some locations display more than 2 responses, highlighting the complexity of identifying the proxy response to the AMOC slowdown or overshoot. Furthermore, for grid cells displaying strictly two responses, the first response does not necessarily correspond to the expected response to the AMOC slowdown. For instance, Figure 2B actually shows a $\delta_{13}$C time series that displays a $\delta_{13}$C increase as first response and a $\delta_{13}$C decrease as a "second" or late response. In this case the first response likely corresponds to an accumulation of nutrients due to the cessation of NADW export as suggested in (Menviel et al., 2015) while the second or late response corresponds to the expected $\delta_{13}$C decrease subsequent of an AMOC slowdown. Therefore, we kept the terminology of "first" and "second" response throughout the manuscript.

The second point of this comment on Figure 3 has also been raised by the reviewer #2 (see response to reviewer #2 comments). We agree with both reviewers; single and dual responses are clearly **mutually exclusive**. However, Figure 3 shows **zonally averaged** proxy responses over the western Atlantic. Consequently, for one grid cell to appear blank on Figure 3 A. (resp. 3.B.) it is required that the zonal average is empty and consequently that there is no grid cell in the full longitude range considered displaying a single response (resp. dual response). Therefore, it is possible to have overlaps on Figure 3. We added a sentence to Figure 3's caption in order to clarify this point:

"Single and dual responses are mutually exclusive on a per location basis. Since panels A and B are showing zonal averages, overlaps may arise from different locations with the same latitude but different longitudes."

2. Responses of Pa/Th and carbon isotopes in the Atlantic in a hosing experiment are not new and have been examined in other studies already. Since this paper focuses on Pa/Th, their modeled Pa/Th response should be compared to previous studies (Gu and Liu, 2017; Rempfer et al., 2017). Spatial and temporal similarities and differences with these previous studies should be compared and discussed.

We thank the reviewer for highlighting that the study from (Gu and Liu, 2017) dealing with Pa and Th in CESM was not cited in the original manuscript. We have now included a citation to this study in the revised manuscript.

It is true that the Pa/Th response to AMOC slowdown in hosing experiments has already been examined in different versions of Bern 3D (Rempfer et al., 2017; Siddall et al., 2007) and in CESM (Gu and Liu, 2017). We agree with the reviewer that it would be of great interest to evaluate the spatial and temporal similarities of the Pa/Th responses in those different models. This is however not an easy task and would require new coordinated modeling experiments with the different models. Indeed, the parameters of the hosing experiments performed (flux, input location and duration) are quite different in our study and in (Gu and Liu, 2017; Rempfer et al., 2017; Siddall et al., 2007). On the novelty of our study, we would like to point out that the Pa/Th response to the AMOC slowdown has not been assessed in a consistent way in the different above cited publications, most of them only displaying the Pa/Th response at selected locations of the North-Atlantic. Likewise, it is worth to point out that our study is the only one to consider spatial and temporal variability of the Pa/Th response to AMOC changes. Therefore, a detailed evaluation of spatial and temporal similarities

as requested by the reviewer is i) not achievable given the existing publications and ii) out of the scope of this manuscript which focuses on the spatial and temporal similarities and differences between 2 carbon isotopes proxies and the Pa/Th in a single model. Given the interest of the topic, we have added a paragraph in the discussion section to acknowledge that Pa/Th response to AMOC slowdown has already been studied in other models and highlight that the Pa/Th response obtained in the present study is quite consistent with what has been observed in previous studies (see revised manuscript).

3. At the end of the introduction, three questions are raised. The first two questions are discussed in section 3 and 4, but the third question "How can the modelled multi-proxy response help to interpret the paleoproxy records" is not clearly answered. The implication for interpreting the paleoproxy records is not clearly state. This is a very important question for modeling proxies in GGMS. Authors need to add some discussion about this kind of implications in the discussion.

We thank the reviewer for this point. One of the main motivations for multi-proxy modelling is to achieve a more efficient model-data comparison by bringing model output closer to the observables (the proxy records). This study is the first one considering $\delta_{13}C$, $_{14}C$ and Pa/Th in a consistent modelling frame and it shows 1) strong spatial variability in the proxy response (according to the main water mass bathing the considered locations) and 2) the possibility for a time delay between proxy responses at a given location (200 year lag of the carbon isotopes response relative to the Pa/Th response in the deep Northwest Atlantic). Therefore, our results show that the interpretation of the proxy data might be complicated because a given circulation change event does not necessarily produce a single and consistent proxy response in the entire Atlantic basin, nor a synchronous multi-proxy response at a given core location. We have now revised the entire discussion and conclusions sections of the manuscript to account for the comments that we received, and we hope the implication of our study for the interpretation of the proxy records is now clearer regarding this topic.

4. More differences between Pa/Th and carbon isotopes in reconstructing past AMOC could be discussed and highlighted. As mentioned above, the novelty of this paper is studying the Pa/Th together with carbon isotopes since the Pa/Th and carbon isotopes in a hosing experiment have been presented in previous studies. However, I feel this multi-proxy comparison is not fully developed in the current manuscript. A more in-depth comparison between Pa/Th and carbon isotopes and their implications for paleoceanography (back to comment 3) are needed.

As already mentioned previously, and further explained below in this response to the reviewers, the discussion section of the manuscript has undergone very substantial revisions. It now includes a more in depth comparison of Pa/Th and carbon isotopes as requested by the reviewer. See for example lines 27 to 40 p8.

5. Pa/Th leads carbon isotopes, but lead by how many years? From Figure 5 a and c, it seems that the 300 years hosing is too short for carbon isotopes to fully adjust to the reduction of AMOC. If hosing is kept longer than 300 years, carbon isotopes may lag Pa/Th response even longer. Therefore, from this 300-years hosing experiment, we cannot determine the exact lead time. This should be pointed out.

As shown on Figure 4, the actual response times and therefore the lag time between Pa/Th and carbon isotopes responses has a strong spatial variability (with locations showing actually no lag, as shown on Figure 5). We agree that 300 years of freshwater addition is likely too short for the carbon isotopes to fully adjust. The revised discussion section now highlights these 2 points.

6. The modeled Pa/Th is compared to observations in Dutay et al., 2009 and Henderson et al., 1999 (Page 5, line 14). However, in recent years, many new observations are now available. GEOTRACES offers a lot of relevant new data (also used in Rempfer et al., 2017). More core top Pa/Th are also available. A more complete compilation of the observations should be used to tune the model parameters. Also, if comparing to the same compilation of observations as in previous studies (Gu and Liu, 2017; Rempfer et al., 2017; Van Hulten et al., 2018), model performance in simulating Pa/Th can be estimated quantitatively.

We agree with the reviewer that the GEOTRACES intermediate data product 2017 (Schlitzer et al., 2018) offers relevant new data. In fact, the core-top data used in Figure 1 is actually the same that was compiled in (van Hulten et al., 2018). We have corrected the citations in the manuscript accordingly. Besides, we have updated the supplementary figures for the sake of a better model performance evaluation. The main and supplementary figures now display: i) the zonally averaged Atlantic dissolved and particulate Pa, Th and Pa/Th (as suggested by the reviewer #2), ii) the model-data comparison along GEOTRACES transect GA03 and GA02S as shown in (Gu and Liu, 2017; van Hulten et al., 2018; Rempfer et al., 2017). The question of assessing the model performance using the GEOTRACES data and comparison with previous studies is developed in the response to reviewer #2- major comment n°1 below.

Minor Comments: 1. Page 3, Line 27, Gu et al. (2017) simulating Pa/Th in CESM should be mentioned here (higher resolution then Rempfer et al. (2017) and longer integration than van Hulten et al. (2018)). Gu, S., Liu, Z., 2017. 231Pa and 230Th in the ocean model of the Community Earth System Model (CESM1.3 ). Geosci. Model Dev. 10, 4723–4742. https://doi.org/10.5194/gmd-10-4723-2017

We thank the reviewer for highlighting this relevant reference, which has been added to the text as suggested (see revised manuscript).

2. Page 4, Authors follow Rempfer et al. (2017) to implement Pa/Th. One advance in Rempfer et al. (2017) in simulating Pa/Th is that bottom scavenging and boundary scavenging are included, which improves the simulation of water column Pa and Th activity. In page 8, line 37, authors state that the bottom and boundary scavenging

are not modeled in LOVECLIM. This should be mentioned earlier in section 2.1 (model description and developments). Also, the modeling scheme (similarities and differences) comparing with previous modeling efforts should be discussed explicitly in section 2.1.

We agree with the reviewer, among the models able to simulate the evolution of the Pa and the Th, Bern 3D is the only one having an explicit parametrization for bottom and boundary scavenging. As stated by the authors, this parametrization is rather crude and consists in scaling (increasing) the Pa scavenging coefficients in the coastal grid-cells in order to achieve enhanced Pa removal at the ocean boundaries (Rempfer et al., 2017).

We would like to point out that all models actually represent the so-called boundary scavenging effect. Indeed, the particles fluxes produced by the GCM NEMO-PISCES and used in iLOVECLIM show greater fluxes at the continental margins compared to the ocean interior. Therefore, the higher particle fluxes induce a greater Pa removal in the regions of high particle fluxes, even in the absence of additional parametrization of the boundary scavenging. Therefore, the need for an additional parametrization of the boundary scavenging does not appear to be fundamental.

The scavenging scheme and modelling choices are fully described in the method section. To date, the Pa and Th have been implemented in at least 5 models of intermediate complexity or GCMs. Therefore, a full discussion of the similarities and differences between these models would represent a model intercomparison project, which is out of the scope of this paper. Nevertheless, we have followed the reviewer's recommendations and modified the method description to i) mention that no explicit parametrization of boundary and bottom scavenging have been included in iLOVECLIM and ii) add information about how the modelling scheme used in this study compares with previous Pa/Th modelling work (see the method section of the revised manuscript).

3. Page 5, Line 20 Details about the PI forcing should be provided. From Figure 5, there is interannual variability. Is the PI forcing looping in the first 300 years?

The meaning of the question from the reviewer is unclear to us. Is the question related to interannual variability in the climate model itself? The setup we are using is a fully coupled atmosphere – ocean – vegetation climate model. Within that climate model system there is some interannual variability simulated in the climate by itself. Regarding the boundary conditions of the climate model, these are fixed to pre-industrial conditions and as such, there is no looping condition. The interannual variability in the model is not a product of the boundary conditions imposed but of the interactions within the atmosphere – ocean – vegetation climate numerical system used.

4. Section 3.1 Vertical structures of Pa/Th could also be provided and compared to observations (GEOTRACES transects), such as Figure 2 and Figure 3 in Rempfer et al. 2017. Figure S1 only have particulate and dissolved Pa and Th.

We would like to highlight that (Gu and Liu, 2017; Rempfer et al., 2017) only provide dissolved Pa and Th as well as particulate Pa/Th (i.e. no particulate Pa and Th) along

the GEOTRACES transects GA03 and GA02S. As detailed above, we have included new supplementary figures showing dissolved, particulate Pa, Th and Pa/Th on a N-S Atlantic section as well as along GEOTRACES transect GA03 and GA02S (see response to previous comments and response to reviewer's 2 major comment n°1).

5. Page 6, section 3.2, first paragraph, Figure 5 can be referred here. Then people can see exactly how the fresh water is added and how the AMOC evolves.

Done

6. Page 6, line 14-16, this sentence can be rewritten for easier understanding.

We have split this sentence into two and added a coma (as suggested by the reviewer #2). We hope this technical information is now clearer.

7. Page 7, line 15, Any explanations for the 14C response time difference between the eastern and western basin?

The NADW is stronger in the western basin (western boundary current). The circulation pathways are hence different in the western and eastern Atlantic, both in real life and in the models. The NADW is less active in the eastern basin, which can explain the $_{14}$C response pattern (see revised manuscript p7 L33).

8. Figure 2 gives two examples of the single response and dual response. What is the proxy exactly? Pa/Th? 13C? or 14C? And where is the grid, location and depth? Also, it would be good to add AMOC in this plot for people to follow.

We thank the reviewer for his/her comment. However, we think that the purpose of Figure 2 has been misinterpreted.

Figure 2 has for only purpose to display the theoretical definition of "proxy response" and "proxy response time" as defined in the text whatever the grid cell, actual location and water depth. What is represented is $\delta_{13}$C but the figure would be valid for any time series for any proxy. We have included the AMOC time series in Figure 2 as suggested by the reviewer.

9. Authors use fixed particle fluxes in their hosing experiment. After adding fresh water to the North Atlantic, the particle fluxes will change. Will this particle flux change affect the results of this paper should be discussed.

We agree with the reviewer that any change of the ocean surface conditions (adding freshwater, temperature…) will likely induce particle fluxes changes (i.e. flux intensity and/or composition). In its current version, with fixed particles, iLOVECLIM does not simulate the impact of primary productivity changes on the Pa/Th. Instead, we only simulate the impact of circulation changes, which is of interest in itself. As stated above, we have revised the discussion section of the manuscript and ensured to clearly state and discuss the implication of the use of fixed and prescribed particle fluxes.

10. The conclusions and perspectives can be improved to highlight the major findings. Currently, it is too broad and descriptive.

We have rewritten the conclusion in order to highlight the major findings.

References:

Gu, S. and Liu, Z.: 231 Pa and 230 Th in the ocean model of the Community Earth System Model (CESM1. 3)., Geosci. Model Dev., 10(12), 2017.
van Hulten, M., Dutay, J. C. and Roy-Barman, M.: A global scavenging and circulation ocean model of thorium-230 and protactinium-231 with realistic particle dynamics (NEMO–ProThorP 0.1), Geosci Model Dev Discuss, 2017, 1–32, doi:10.5194/gmd-2017-274, 2018.
Menviel, L., Mouchet, A., Meissner, K. J., Joos, F. and England, M. H.: Impact of oceanic circulation changes on atmospheric $\delta 13CO_2$, Glob. Biogeochem. Cycles, 29(11), 1944–1961, doi:10.1002/2015GB005207, 2015.
Rempfer, J., Stocker, T. F., Joos, F., Lippold, J. and Jaccard, S. L.: New insights into cycling of 231Pa and 230Th in the Atlantic Ocean, Earth Planet. Sci. Lett., 468, 27–37, doi:https://doi.org/10.1016/j.epsl.2017.03.027, 2017.
Schlitzer, R., Anderson, R. F., Dodas, E. M., Lohan, M., Geibert, W., Tagliabue, A., Bowie, A., Jeandel, C., Maldonado, M. T. and Landing, W. M.: The GEOTRACES intermediate data product 2017, Chem. Geol., 493, 210–223, 2018.
Siddall, M., Stocker, T. F., Henderson, G. M., Joos, F., Frank, M., Edwards, N. R., Ritz, S. P. and Müller, S. A.: Modeling the relationship between 231Pa/230Th distribution in North Atlantic sediment and Atlantic meridional overturning circulation, Paleoceanography, 22(2), doi:10.1029/2006PA001358, 2007.

---

## Author Comment (AC2) · 17 Dec 2019

**Carbon isotopes and Pa/Th response to forced circulation changes: a model perspective**

**Note to the Editor and referees**

We have requested a long deadline extension since a bug was found in the iLOVECLIM model code that could affect the carbon cycle part. To ensure scientific reproducibility, we first wanted to assess whether the bug found could have a significant impact on our results. This has required to run several multi-millennial long simulations. After analysis of the results of the newer version, we however found that our conclusions are unaffected by this error. We thank the Editor and the referees for their patience in this necessary process.

**Response to the referees' comments**

We thank both reviewers for their constructive comments that helped to improve and clarify the manuscript. We have addressed the comments in detail below.

**Anonymous Referee #2**

This is a report on the implementation of the Pa/Th sedimentary proxy in the ocean-climate model of intermediate complexity, iLOVECLIM, in addition to the previously included stable carbon and radiocarbon isotope ratios. The reconstruction of past circulation states has suggested substantial changes from that observed in the modern ocean, with potentially significant implications for past climate change. It is therefore important that model simulations can capture the observed sedimentary evidence and demonstrate the ocean physics that might be consistent with this evidence. In this case, the incorporation of multiple isotopic tracers with different distribution and influences adds a valuable layer of sophistication to such modeling efforts.

In addition to demonstrating the model's ability to reproduce the observed modern distributions of Pa/Th and carbon isotopes, the authors report on the results of what is now a relatively standard "hosing" experiment, wherein freshwater is imposed on the surface of the high latitude North Atlantic within the model domain, in order to weaken convection and the overturning circulation. Changes in subsurface water masses and the strength of the overturning have the result of redistributing the sedimentary Pa/Th and carbon isotopes, which the authors then interpret and compare to existing data. They identify different responses of in the respective tracers. One major finding is that in the hosing experiment, changes in both carbon isotopes lag Pa/Th by a few hundred years.

Overall, this study is an important step forward in terms of the state of the art of implementation of circulation proxies and should therefore be worth accepting for publication in Climate of the Past, following revisions that should address the following points.

A more careful data-model comparison is needed to validate the simulated Pa and Th, which is the main advance made in the model. The paper compares bottom water particulate Pa/Th with a core top compilation (Henderson et al., 1999). Other than that, the comparison with Pa/Th data is mostly qualitative. The authors acknowledge that they refrained from making more data/model comparisons because of the crudeness of the model (Page 8 Line 41 (P8L41)). However, it is still important to show those comparisons. Readers may gain information about the fidelity (or lack thereof) of the model to the modern observations, including regions where the model performs well and regions where it does not. This information will help make the audience more informed, and therefore increase the impact of the paper. For example, the paper mentions the compilations of sedimentary Pa/Th by Lippold et al. (2016) and Ng et al. (2018) (P9L6). How does the model compare with them graphically? How do the particulate and dissolved Pa and Th results compare with GEOTRACES observations? These could be addressed in a few brief passages.

The two reviews received highlighted the need for a more careful model data-comparison. We agree that it is important to show where the model performs well and where it does not. We have thus revised the corresponding set of figures (Figures S1 to S3) and text accordingly. As already mentioned, the core top compilation used in Figure 1 is actually the same compilation as presented in (van Hulten et al., 2018), we have corrected the references in the manuscript accordingly. Besides, we are now presenting the relevant modelled variables on the GEOTRACES transects GA03 and GA02S shown in previous studies (Gu and Liu, 2017; van Hulten et al., 2018; Rempfer et al., 2017) as detailed below.

Both reviewers mentioned the need of a more quantitative model evaluation. As explained in the supplementary text S1, we have calculated the RMSE of dissolved and particulate Pa and Th and noticed that while improving the Th variables, we were actually deteriorating the Pa variables and vice versa… Besides, we would like to point out that none of the previously published Pa/Th implementations really achieved a proper quantitative evaluation of the model-data agreement between model PI output and modern datasets. What is usually shown is the dissolved Pa and Th profiles along with the particulate Pa/Th on GEOTRACES transects GA03 and GA02 ((Gu and Liu, 2017; van Hulten et al., 2018; Rempfer et al., 2017)) as well as the sedimentary Pa/Th against Holocene core top data and remains a graphical and qualitative evaluation.

At this stage, and because at least 5 implementations of Pa and Th in different models of intermediate complexity and GCMs have been published so far, a quantitative model-data evaluation would make more sense in the frame of an extensive model-model and model-data intercomparison. Such a work is clearly out of the scope of this study and is the subject of ongoing work by the lead author for a subsequent publication.

In addition, we would like to highlight a few points concerning the use of the GEOTRACES data for model performance evaluation. It is clear that the GEOTRACES database provides a growing amount of data for comparison with model outputs. However, we think that it is worth to point out a few issues:

- First of all, it is worth noting that the model outputs represent averages of several years of run under equilibrium state while GEOTRACES data

correspond to one particular sampling date. Besides, depending on the model resolution, several GEOTRACES data points can correspond to one single model grid box… All in all, it seems important to us to stress that the model can sometimes display a feature "at the wrong" position, so any point by point comparison has to be handled carefully.

- What is reported in the GEOTRACES dataset (Schlitzer et al., 2018) are total dissolved or particulate $_{231}$Pa and $_{230}$Th activities corrected for measurement blanks and ingrowth since sample collection. These relatively "raw" concentrations do contain a signal from the $_{231}$Pa and $_{230}$Th coming from detrital (dissolution of terrigenous material or terrigenous component of particles) as well as $_{231}$Pa and $_{230}$Th coming from water column scavenging (also called excess fraction). The different fractions/contributions can be derived from the $_{231}$Pa, $_{230}$Th and $_{232}$Th concentrations using a few assumptions and can sometimes represent more than 10% of the signal. It seems important to remind that what is actually computed by the models is solely the Pa and Th derived from the water column U-decay. Any other source of Pa and Th is not taken into account by the models. However, none of the published modelling paper mentions which concentrations (corrected or not – which correction) have been considered. This complicates the evaluation of the model performances and motivates an extensive model intercomparison. For the reasons mentioned above, using the full potential of the GEOTRACES dataset would therefore require to first determine a method for the calculation of the excess fractions as the pre-calculated excess concentrations are only available for 2 Atlantic profiles. Such work is clearly beyond the scope of the present study.

For all of the reasons explained above and for the sake of consistency with the model evaluations that have been previously published we are now showing for the model-data evaluation:

- The modelled sedimentary Pa/Th against the core top database as shown in (van Hulten et al., 2018) – see response to reviewer 1 comments and the revised caption of Fig. 1
- The zonally averaged dissolved and particulate Pa, Th and Pa/Th N-S Atlantic profiles (as shown on Figure 6 (Gu and Liu, 2017)) – Figure S1.
- The dissolved Pa and Th along GEOTRACES transect GA02 S using the data from (Deng et al., 2014) – please note that there are no particulate (only dissolved) data published in the latter article – Fig. S2
- The dissolved and particulate Pa and Th and Pa/Th along the GEOTRACES transect GA03 using the data from (Hayes et al., 2015a, 2015b) – Fig. S3

The model-data agreement of the water column particulate and dissolved activities is extensively described in the SOM.

Additionally, hypotheses are offered for why 13C response leads Pa/Th (possibly biology and/or air-sea exchange slows down 13C response), yet given the setup of the model, it would be a missed opportunity not to conduct a more detailed diagnosis of modeled causes for the lead-lag relationships among the various tracers. If the reasons can be pinned down, the paper can make a more robust conclusion, even if it is model-depending. Is it possible to plot the biological changes before and after a hosing experiment? How about changes in the air-sea exchange rate? Depending on the

results of those plots, the paper can then present a fuller picture of the changes during a hosing experiment.

We thank the reviewer for asking us to further investigate the lag between carbon isotopes and Pa/Th responses to a decrease in NADW formation. Indeed, the model set up allows us to look at the total biologic productivity (Calcium carbonate and particulate organic carbon) changes across the hosing. However, as shown on Figures 4 and 5, the carbon isotopes response has a strong spatial variability. Therefore, plotting the marine productivity anomaly between the control period and the hosing peak (year 550 to 600) – i.e. before and after the hosing as suggested by the reviewer – will not help investigating the cause of the delayed carbon response which happens around year 800 in the deep NW Atlantic.

In order to investigate the potential causes for the carbon isotopes lag, we have plotted time series of relative organic carbon production, aqueous $pCO_2$ and sea surface temperature averaged over the North Atlantic, where the lag between Pa/Th and carbon isotopes is the most pronounced. We see that, in line with the classical hosing response described in the literature, the freshwater addition in the North Atlantic causes a decrease of the SST and the organic carbon production as well as an increase of the aqueous $pCO_2$. However, all those changes reach their maximum around year 600 (or shortly before) of the simulation, i.e. about 200 years before the $\delta_{13}C$ response around year 800 (in the deep basin). This indicates that the air-sea exchanges and the biological productivity are not directly responsible for the time lag between the carbon isotopes response and the AMOC slowdown. Looking at a series of N-S Atlantic sections, we can see that the $\delta_{13}C$ anomaly builds up in the North Atlantic intermediate waters above 3000 m from year 400 to 750. The maximum of the anomaly is located between 1500 and 3500 m around 50°N and spreads in the deep Atlantic from year 750. Therefore, we argue that the lag between the carbon isotopes response relative to the Pa/Th is likely due to the time necessary to transport the anomaly on site, in particular in the deep ocean. While the Pa/Th directly depends on the AMOC capacity to transport Pa southwards, the carbon isotopes form in the intermediate ocean because of changes in air-sea exchange and accumulation of nutrients related to the decrease in NADW formation. Therefore, it seems to take more time for the carbon isotopes anomaly to reach the deeper ocean through water mass advection.

We have revised the discussion section and added Figure 6 to include those findings in the revised manuscript. This study highlights a complex response of the different proxies to a rather classic circulation perturbation. The relationship between the proxies and the ocean circulation surely needs further work to be fully understood. We hope that our manuscript now presents a fuller picture of the changes happening during a hosing experiment.

Additional smaller points for consideration:
In P1L25 This is confusing. Should it read "without an a priori guess"?

Yes, we have now corrected this sentence.

In P1L33-34 These are not global changes and should be more narrowly defined, possibly as regional or even local.

Done

In P1L36, it should be "see Lynch-Stieglitz (2017) for a review".

Done

In P1L40, "Nd, Cd/Ca, sortable silt are also valuable proxies to reconstruct circulation and water mass and they are worth mentioning.

We have now changed the sentence by:

"To date, among the numerous tracers available (e.g. benthic $\delta^{18}O$, Nd isotopes, Cd/Ca or sortable silts), the sedimentary ($^{231}Pa_{xs,0}/^{230}Th_{xs,0}$) ratio (hereafter Pa/Th) and dissolved inorganic carbon isotopes ($\delta^{13}C$, $\Delta^{14}C$) are key tracers to reconstruct and quantify past circulation patterns and water mass flow rates."

In P2L25, this should be 12C, although in truth it is both, with a lower 13C/13C.

We thank the reviewer for highlighting this typo. This has now been corrected.

In P4L17, Equation 2, the second minus sign is different from the first. The multiplication dot is positioned as a punctuation would.

This has now been corrected

In P4L34, Equation 4, the "d" in "Kd" should be subscript.

Done

In P5L14, it should be "compiled in Dutay et al. (2009)".

This comment is no longer valid as we now evaluate the model performance using GEOTRACES dataset as requested by the two reviewers.

In P6L13, this is a long and potentially confusing sentence, yet a valuable one for its description of how the proxies were evaluated. It would help to have a comma after "identify", which might make it clearer that the identification is of simulation periods exceeding a defined length, for each proxy.

We thank the reviewer for the suggestion that we adopted.

In P8L3-5, this is a bold statement that just does not ring true. Ten thousand years for equilibration of the carbon isotope signal in the Atlantic ocean, and a thousand for Pa/Th? This is in a basin where the residence time of the deep waters is a few hundred years today, and maybe a thousand years more in the past. Unless I misunderstand the point here, something is not right.

We thank the reviewer for his/her careful reading. However, we confirm that this is not a "bold wrong statement". Indeed, because Pa and Th, on the one hand, and the carbon isotopes, on the other hand, undergo very different processes in the ocean water column, the simulation time required to equilibrate these different tracers is very different. For instance, carbon isotopes are exchanged between different reservoirs such as the atmosphere, the ocean, the terrestrial and marine biosphere while the Pa and Th stay in the oceanic compartment.

We indeed agree that the residence time of deep waters is about a few hundreds of years today and could have increased up to thousands of years in the past in the Atlantic basin. However, Pa and Th residence time in the water column is **not related** to the residence time of the deep-water masses. Instead, the residence time of Pa and Th in the water column depends on how fast these two isotopes are scavenged to the sediments by particles sinking in the water column. In other words, the **observed** residence time of Pa and Th in the water column remains up to **200 years for Pa** and **40 years for Th** (Henderson and Anderson, 2003), independently of the overturning circulation rate.

From the model side, (van Hulten et al., 2018) obtain negligible Pa and Th drifts, of less than 0.1% after 500 years of equilibration. Please find below the text from (van Hulten et al., 2018) on this topic :

*"The model was spun up for 500 years, after which it was in an approximate steady state (decadal drift of −0.002 % for total $_{230}$Th and +0.058 % for total $_{231}$Pa). Protactinium-231 has a larger drift than thorium-230 because $_{230}$Th is more quickly removed everywhere in the ocean because of its high particle reactivity. The lithogenic particles are in a steady state, and the PISCES variables are in an approximate steady state (e.g. phosphate shows a drift of −0.005 % per decade)."*

The figure below shows the evolution of the global averages for dissolved and particulate Pa and Th after a fresh-start (all activities are initialized to 0). Therefore, we may consider that the Pa/Th tracer is at equilibrium after 1000 years of simulation.

[Figure]

In P8L26, when listing the reasons that the millennial scale climate changes are not analogous to the hosing experiments, it would be useful to also point out that the location of the freshwater hosing in the model (the Nordic Seas) could also be different from events that originate primarily from the Laurentide ice sheet on North American, mostly likely including the millennial Dansgaard-Oeschger and Heinrich events.

We thank the reviewer for highlighting this point. We have changed the text accordingly:

"However, our hosing experiments are not direct analogues of the millennial scale climate changes of the last glacial cycle because i) glacial millennial events occurred under glacial conditions whereas our simulations were run under PI conditions, ii) the Heinrich and DO events have distinct proxy patterns and cannot be entirely explained by a simple fresh water addition in the North Atlantic and iii) the freshwater inputs might have occurred in different locations across distinct millennial scale events (e.g. originating from the Laurentide or Scandinavian ice sheet) while in the model, the freshwater was only added in the Nordic seas."

P9L10, substitute "If" with "While".

Done

P9L32, modeled 13C results are compared with another model's results, yet this section is named "Comparison to proxy data". Maybe update section title to "Comparison to proxy and modeled data."

We thank the reviewer for his suggestion that we have implemented.

In Figure 2, the labels (e.g. A. Single response) are unnecessarily far from the plots. The caption should define the dotted black lines (which I assume is the 2 sigma variation of the control phase). Also in the caption, the dotted red vertical line is the response time and the dotted red horizontal line is the proxy response. The caption states it the other way.

We thank the reviewer for highlighting these issues. Figure 2 and its caption have been edited accordingly.

In Figure 3, I think I'm missing something here. Why are there overlaps between the data coverage of single and dual response plots? Shouldn't the two be mutually exclusive?

As pointed out by the reviewer, single and dual responses are clearly mutually exclusive. However, as explained in our response to reviewer 1, Figure 3 shows zonal averages for the western Atlantic N-S section. Consequently, for one grid cell to appear blank on Figure 3 A. (resp. 3.B.) it is required that the zonal average is empty and consequently that there is no grid cell in the full longitude range considered, displaying a single response (resp. dual response). Therefore, it is possible to have overlaps on Figure 3. We added a sentence to Figure 3's caption in order to clarify this point.

Lastly, a citation in your references has the wrong publication year. The citation for "Luo, Y., Francois, R. and Allen, S. E.: Sediment 231Pa/230Th as a recorder of the rate of the Atlantic meridional overturning circulation: insights from a 2-D model., Ocean Science Discussions, 6(4), 2755–2829, 2009." should instead be Luo, Y., Francois, R., and Allen, S. E.: Sediment 231Pa/230Th as a recorder of the rate of the Atlantic meridional overturning circulation: insights from a 2-D model, Ocean Sci., 6, 381-400 https://doi.org/10.5194/os-6-381-2010, 2010.

We thank the reviewer for highlighting this typo. This is now corrected.

References:

Deng, F., Thomas, A. L., Rijkenberg, M. J. A. and Henderson, G. M.: Controls on seawater 231Pa, 230Th and 232Th concentrations along the flow paths of deep waters in the Southwest Atlantic, Earth Planet. Sci. Lett., 390, 93–102, doi:10.1016/j.epsl.2013.12.038, 2014.
Gu, S. and Liu, Z.: 231 Pa and 230 Th in the ocean model of the Community Earth System Model (CESM1. 3)., Geosci. Model Dev., 10(12), 2017.
Hayes, C. T., Anderson, R. F., Fleisher, M. Q., Huang, K.-F., Robinson, L. F., Lu, Y., Cheng, H., Edwards, R. L. and Moran, S. B.: 230Th and 231Pa on GEOTRACES GA03, the U.S. GEOTRACES North Atlantic transect, and implications for modern and paleoceanographic chemical fluxes, Deep Sea Res. Part II Top. Stud. Oceanogr., 116, 29–41, doi:https://doi.org/10.1016/j.dsr2.2014.07.007, 2015a.
Hayes, C. T., Anderson, R. F., Fleisher, M. Q., Vivancos, S. M., Lam, P. J., Ohnemus, D. C., Huang, K.-F., Robinson, L. F., Lu, Y., Cheng, H., Edwards, R. L. and Moran, S. B.: Intensity of Th and Pa scavenging partitioned by particle chemistry in the North Atlantic Ocean, Mar. Chem., 170, 49–60, doi:https://doi.org/10.1016/j.marchem.2015.01.006, 2015b.
Henderson, G. M. and Anderson, R. F.: The U-series toolbox for paleoceanography, Rev. Mineral. Geochem., 52(1), 493–531, doi:10.2113/0520493, 2003.
van Hulten, M., Dutay, J. C. and Roy-Barman, M.: A global scavenging and circulation ocean model of thorium-230 and protactinium-231 with realistic particle dynamics (NEMO–ProThorP 0.1), Geosci Model Dev Discuss, 2017, 1–32, doi:10.5194/gmd-2017-274, 2018.
Rempfer, J., Stocker, T. F., Joos, F., Lippold, J. and Jaccard, S. L.: New insights into cycling of 231Pa and 230Th in the Atlantic Ocean, Earth Planet. Sci. Lett., 468, 27–37, doi:https://doi.org/10.1016/j.epsl.2017.03.027, 2017.
Schlitzer, R., Anderson, R. F., Dodas, E. M., Lohan, M., Geibert, W., Tagliabue, A., Bowie, A., Jeandel, C., Maldonado, M. T. and Landing, W. M.: The GEOTRACES intermediate data product 2017, Chem. Geol., 493, 210–223, 2018.

---

## Editor Decision (ED1)

**Carbon isotopes and Pa/Th response to forced circulation changes: a model perspective**

L. Missiaen[1,2], N. Bouttes[1], D.M. Roche[1,3], J-C. Dutay[1], A. Quiquet[1,4], C. Waelbroeck[1], S. Pichat[5,6], J-Y 
[revised manuscript text omitted]

| | σPa-CaCO3 | σPa-POC | σPa-opal | σTh-CaCO3 | σTh-POC | σTh-opal |
|---|---|---|---|---|---|---|
| Best fit | 1.87 | 1.55 | 7.62 | 76.83 | 5.47 | 3.77 |
| | KdPa-CaCO3 | KdPa-POC | KdPa-opal | KdTh-CaCO3 | KdTh-POC | KdTh-opal |
| Best fit | 8.01E+06 | 5.53E+07 | 4.86E+07 | 3.29E+08 | 1.96E+08 | 2.40E+07 |